

# Preferential protein depolymerization as a preservation mechanism for vascular litter decomposing in *Sphagnum* peat

Hendrik Reuter[1], Julia Gensel[2], Marcus Elvert[2], and Dominik Zak[3]

[1]Department of Chemical Analytics and Biogeochemistry, Leibniz-Institute of Freshwater Ecology and Inland Fisheries, DE-12587 Berlin, Germany
[2]MARUM - Center for Marine Environmental Sciences and Faculty of Geosciences, University of Bremen, DE-28359 Bremen, Germany
[3]Department of Bioscience, University of Aarhus, DK-8600 Silkeborg, Denmark

**Correspondence:** Hendrik Reuter (hendrik.reuter@mail.de)

**Abstract.** Nitrogen (N) dynamics in *Phragmites australis* litter due to anaerobic decomposition in three anoxic wetland substrates were analyzed by elemental analyses and infrared spectroscopy (FTIR). After 75 days of decomposition, a relative accumulation of bulk N was detected in most litters, but N accumulated less when decomposition took place in a more N-poor environment. FTIR was used to quantify the relative content of proteins in litter tissue and revealed a highly linear relationship

between bulk N content and protein content. Changes in bulk N content thus paralleled and probably were governed by changes in litter protein content. Such changes are the result of two competing processes within decomposing litter: enzymatic protein depolymerization as a part of the litter breakdown process and microbial protein synthesis as a part of microbial biomass growth within the litter. Assuming microbial homeostasis, DNA signals in FTIR spectra were used to calculate the amount of microbial N in decomposed litter which ranged from 14 to 42% of the total litter N for all leaf samples. Microbial carbon

(C) content and resultant calculated carbon-use efficiencies (CUEs) indicate that microbial N in litter accumulated according to predictions of the stoichiometric decomposition theory. Subtracting microbial C- and N-contributions from litter, however, revealed decomposition site dependent variations in the percentual amount of remaining, still unprocessed plant N compared to remaining plant C, an indicator for preferential protein depolymerization. For all leaf litters, the coefficient of preferential protein depolymerization ($\alpha$), which relates N-compound depolymerization to C-compound depolymerization, ranged from

0.74 - 0.88 in a nutrient-rich detritus mud to 1.38 - 1.82 in *Sphagnum* peat, the most nutrient-poor substrate in this experiment. Preferential protein depolymerization leads to a gradual N depletion of decomposing litter which we propose as a preservation mechanism for vascular litter decomposing in *Sphagnum* peat.

## 1 Introduction

Accounting for 2-15% tissue dry mass, proteins are a minor biopolymeric constituent of plant litter (Kögel-Knabner, 2002). Yet,

protein decomposition, commonly also called gross protein depolymerization, is a decisive sub-process of litter decomposition because proteins contain about 60% of the total litter nitrogen (N), an essential nutrient for all organisms (Geisseler et al., 2010; Schulten and Schnitzer, 1997). Protein depolymerization is the process by which proteins in dead organic matter are



broken down into smaller, N-containing fragments (amino acids) which are readily absorbable by microbial decomposers. Protein depolymerization is often described as a central driver of the terrestrial N cycle as it is the dominant process by which chemically bonded N, inaccessibly stored in litter or soil, becomes accessible for microorganisms or higher plants (Schimel and Bennett, 2004; Wild et al., 2015; Mooshammer et al., 2014a).

Despite its relevance, protein depolymerization is rarely discussed in litter decomposition studies, what can be attributed to analytical constraints in quantifying this process (Schimel and Bennett, 2004). Yet, protein depolymerization and its role in N cycling is not fundamentally distinct when compared to the depolymerization of "carbon"-biopolymers, like lignin or cellulose, and their role in C cycling. Using a simplified picture, microbial decomposers release exoenzymes into their environment, which depolymerize the plant biopolymers in litter (Figure 1). The thereby liberated monomers (sugars, fatty acids, amino

acids, etc.) are assimilated by microorganisms and are either oxidized to generate metabolic energy or used as substrates for microbial biomass growth. While in the first case, the depolymerized organic matter is lost from the litter, the fraction of depolymerized organic matter used for growth remains within the litter as newly formed microbial biomass (Reddy and DeLaune, 2008). Concerning C-cycling and N-cycling during litter decomposition, the critical difference between the two can be traced back to the fact, that C losses from litter are acceptable measures to decribe litter decomposition, because the

amount of depolymerized C converted into microbial biomass is comparably small (Schimel and Bennett, 2004; Sinsabaugh et al., 2013; Manzoni et al., 2012). A switch from litter C loss or mass loss, which are mineralization-based measures, to C-compound depolymerization is consequently not required to study C dynamics during litter decomposition.

For N-cycling, the corresponding processes are protein depolymerization (or more generally N-compound depolymerization) and N mineralization, which is the breakdown of amino acids into ammonium ($NH_4^+$), $CO_2$ and $H_2O$ (Schimel and Bennett,

2004; Miltner and Zech, 1999; Andresen et al., 2015). Yet, litter N losses or $NH_4^+$-formation are inadequate tools to estimate protein depolymerization. Microbial biomass has a protein content of 40-60% dry mass and the microbial N demand for growth and activity is correspondingly high (Naumann, 2006). The N content in litter, on the other hand, can be very variable and is often low. Microbes must not sustain N mineralization for metabolic energy generation as they do for C. They simply use as much N as they need for their metabolic activity and only release N-surpluses to the environment. The nitrogen-use

efficiency (NUE) is consequently more variable than the CUE, with measured values ranging from 15%, if the substate is rich in proteins, to 100% if protein-poor litter is being decomposed. In the latter case, all N released from plant tissue due to protein depolymerization is converted to microbial biomass N. The net N loss to the environment is zero and the relative litter N content sharply increases as C is constantly lost (Mooshammer et al., 2014a). Bulk N losses during litter decomposition are consequently no useful indicators on how much plant protein is being depolymerized, or has been depolymerized at a

certain stage of litter decomposition. To assess N cycling during litter decomposition, there is consequently a need to quantify depolymerized protein in decomposed litter. Yet, the available methods to do so are limited as it is challenging to differentiate the two N fractions in decomposed litter, which are the still unprocessed plant N and newly formed microbial N (Tremblay and Benner, 2006).

The lack of analytical methods to quantify protein depolymerization furthermore sets a limit on the study of the effects of

external N on litter decomposition. In addition to obtain N through protein depolymerization, microorganisms are capable of





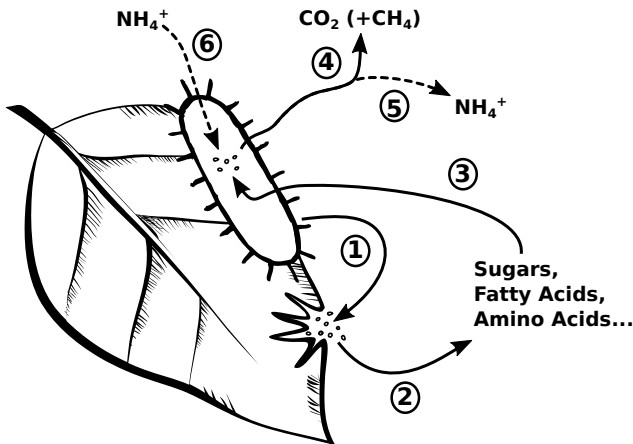

**Figure 1.** A conceptual view on C and N cycling during litter decomposition. Microbial decomposers release exoenzymes (1), which depolymerize plant biopolymers to soluble monomers (2). The relative distribution of formed monomers depends on plant tissue composition and specific enzyme activities. Microbial decomposers assimilate the monomers (3) and either use the contained organic matter for growth or to obtain energy (4). Microbial growth leads to an accumulation of microbial biomass within litter. N mineralization leads to a loss of ammonium to the environment (5), but this process is only active if amino acid-N supply (due to protein depolymerization) exceeds microbial N demand. Ammonium from the environment can additionally serve to fulfill microbial N demand (6).

using $NH_4^+$ from the environment to fullfill their N demand, a process called N immobilization (Schimel and Bennett, 2004). Direct evidence for N immobilization is commonly found when N-poor litter decomposes in which case the total amount of N in litter can exceed the N amount initially present in the undecomposed litter (net N immobilization) (Tremblay and Benner, 2006). N immobilization can in this case accelerate litter decomposition as an otherwise occuring N limitation, which might

limit microbial growth and activity, is prevented (Gulis and Suberkropp, 2003; Rejmánková and Houdková, 2006). Effects of exogenous N on litter N dynamics during decomposition are often found (Prescott, 2005; Moore et al., 2006) and are of high scientific interest due to the potential impact of antrophogenic nutrient inputs on C and N cycling on an ecosystem scale. Results of litter decomposition experiments, however, report contrasting effects. Site N enrichment can inrease N accumulation in litter and decomposition rates (Hobbie and Gough, 2004; Gulis and Suberkropp, 2003; Rejmánková and Houdková, 2006),

while other studies found no effect of N fertilization (Keuskamp et al., 2015) or even decreasing decomposition rates (Luo et al., 2018; Ågren et al., 2001). When vascular litter decomposes in *Sphagnum* peat, unexpectedly high net N losses from litter can occur (Scheffer et al., 2001; Verhoeven et al., 1990). Traditionally, such site-dependent variations in net N mineralization or immobilization are, slightly arbitrarily, accounted for variations in the accumulation of microbial N. Variations in the amount of remaining, unprocessed plant N as a result of different protein depolymerization rates are rarely discussed as a potential

underlying mechanism.

In this study, we report about a litter decomposition experiment we performed in anoxic wetland substrates of three peatlands of different biogeochemical characteristics. As decomposing litter, we chose *Phragmites australis* leaves and rhizomes from the





three sites which spanned a strong gradient in initial N content. To disentangle litter compositional effects and decomposition site effects, leaf litter from each site was decomposed in each organic soil substrate. Although the characterization of litter decomposition in pristine and disturbed peatlands was the initial objective, we in particular found site-dependent variations of net N mineralization/immobilization in litter. To elucidate the reason for these patterns, we developed a new method to

disentangle plant and microbial N in decomposed litter using infrared spectroscopy (FTIR). Our results indicate, that site-dependent deviation of N dynamics were not due to variation in the accumulation of microbial N within litter, but due to variations in the remaining amount of plant N, an indicator of preferential protein depolymerization.

## 2   Materials and methods

### 2.1   Decomposition Experiment

To study N cycling in decomposing litter we performed a 75-day litterbag experiment. *P. australis* tissues and anoxic soil substrates were collected from three peatlands in Northeast Germany. Anoxic soil substrates were collected in June 2013 and included *Sphagnum* peat from an oligotrophic kettle-hole mire (peat C/N = 38.6), sedge-brown moss peat from a mesotrophic terrestrialisation mire (peat C/N = 23.1), and a detritus mud from an eutrophic rewetted fen (sediment C/N = 14.4). Further site characteristics are presented in Table S1. All organic substrates were filled in polyethylene containers (80 x 60 x 43 cm), water

from the corresponding site was added to maintain waterlogged conditions, and the containers were transported to the lab three days before starting the experiment. The containers were stored in the dark at room temperature during the whole experimental time.

*P. australis* leaf litter was collected from the same three sites in autumn 2012, air-dried and stored at -20 °C until start of the experiment. We only collected brown leaves that were still connected to the plant. Leaves were unfreezed shortly before starting

the experiment. To minimize abiotic leaching in the initial phase of decomposition, all leaves were subject to a pre-leaching procedure. In detail, subsets of 25 g leaves were leached in 5 L 1.5 mM NaCl solution, prepared from ultrapure water, for 17 h with one water exchange after 14 h. The leachate was discharged, the leaves were freeze-dried and cut into pieces of 5 cm lenght, thereby discarding leaf base and apex. During soil sampling, *P. australis* rhizome litter was harvested from 10 to 30 cm below the surface at all three sites. The rhizomes were directly freeze-dried and cut into pieces of 5 cm length.

Triplicate litterbags (Nylon, 8 x 8 cm, 0.5 mm mesh size), filled with 1 g preleached leaves or 3 g rhizomes, were incubated close to the bottom of each soil container. Leaves from each site were incubated in each soil substrate, rhizomes only in their home substrate. All litterbags were recovered after 75 days, litters were gently rinsed with distilled water and freeze-dried. After weighing, litters were milled (Pulverisette 9, Fritsch GmbH) and stored at -20 °C in amber glass vials. Three undecomposed subsamples of each litter type were analyzed accordingly to the decomposed samples.






## 2.2 Chemical Analyses

The C and N content of milled samples was measured using an element analyzer (Vario EL by Elementar). Porewater analyses were performed according to Zak et al. (2015) and data is presented in Table S1.

## 2.3 Infrared Spectroscopy

FTIR spectra of initial and decomposed litter samples were measured in transmittance mode using the KBr pellet technique. 1 mg finely milled sample was diluted with 300 mg oven dried KBr (for IR spectroscopy, MERCK) and homogenized with a sample mill. The sample mixture was transferred to an evacuable KBr pellet die, evacuated for 1 min and pressed at 20,000 psi for 4 min to form KBr pellets with a diameter of 13 mm and about 1 mm thickness. FTIR spectra were recorded at a spectral resolution of $4\,cm^{-1}$ and 200 accumulated scans with a Shimadzu IRTracer-100 spectrophotometer, equipped with a DLaTGS detector. Backgroud scans were analyzed before each sample run with pellets of pure KBr.

Using the LabSolution IR software (Shimadzu Corp.), the FTIR spectra were converted into absorbance units and baseline corrected. Second derivative spectra were calculated with the Savitzky-Golay method with 13 convolution points. After data export, all absorbance and second derivative spectra were vector-normalized in the spectral range 1900 to $800\,cm^{-1}$ using Microsoft Excel (Version 12.3.6). FTIR difference spectra were calculated by spectral subtraction of specific vector normalized spectra with the average of the spectra of the three replicate undecomposed samples of the same litter type.

A curve fitting was performed in the region 1850 to $1300\,cm^{-1}$ using OriginPro Peak Analyzer software (Version 8.5.0) using Voigt-shaped bands. To achieve reproducible fitting results, constraints were imposed on Gaussian and Lorentzian peak widths and positions, leaving the peak height as the only parameter to be optimized by the software. Before fitting, the baseline was set to zero at $1900\,cm^{-1}$, what in addition required a renormalization of spectra. Information on peak widths and position can be found in Table S2.

## 2.4 Statistical analyses

All the statistical analyses were performed using the R software "RStudio" (Version 1.1.383, RStudio, Inc.). To test differences of parameters the Mann-Whitney-U test (Wilcoxon rank sum test in R) was performed and for testing dependencies the Kruskal-Wallis rank sum test in combination with pairwise comparisons using the Wilcoxon rank sum test was performed. The relationsship between amide I, amide II, and bulk litter N content were tested with combined data of all initial and decomposed plant litters using the linear regression model of R.





**Table 1.** Bulk C and N content and C/N-values of initial and anaerobically decomposed *Phragmites australis* leaves and rhizomes.

| Decomp. Environment and Litter Type | $N_{initial}$ | $N_{final}$ | $C/N_{initial}$ | $C/N_{final}$ | $C_{loss}$ | $N_{loss}$ |
|---|---|---|---|---|---|---|
| | (wt%) | (wt%) | (atom) | (atom) | (%) | (%) |
| **— high-N substrate —** | | | | | | |
| (detritus mud) | | | | | | |
| low-N leaf | $0.96 \pm 0.03$ | $1.71 \pm 0.03$ | $53.3 \pm 2.4$ | $30.2 \pm 0.2$ | $44.0 \pm 1.7$ | $1.4 \pm 3.6$ |
| medium-N leaf* | $1.39 \pm 0.07$ | $1.97 \pm 0.13$ | $35.5 \pm 0.9$ | $26.5 \pm 1.3$ | $33.9 \pm 1.5$ | $11.6 \pm 4.5$ |
| high-N leaf | $2.14 \pm 0.11$ | $2.83 \pm 0.09$ | $25.5 \pm 1.7$ | $19.9 \pm 0.5$ | $40.2 \pm 1.9$ | $23.5 \pm 1.6$ |
| **— medium-N substrate —** | | | | | | |
| (sedge-brown moss peat) | | | | | | |
| low-N leaf* | $0.96 \pm 0.03$ | $1.67 \pm 0.05$ | $53.3 \pm 2.4$ | $30.4 \pm 0.8$ | $45.0 \pm 2.4$ | $3.8 \pm 4.3$ |
| medium-N leaf | $1.39 \pm 0.07$ | $1.84 \pm 0.03$ | $35.5 \pm 0.9$ | $27.2 \pm 0.5$ | $38.7 \pm 0.5$ | $20.3 \pm 2.0$ |
| high-N leaf | $2.14 \pm 0.11$ | $2.66 \pm 0.16$ | $25.5 \pm 1.7$ | $21.7 \pm 1.1$ | $43.0 \pm 2.2$ | $33.3 \pm 0.7$ |
| **— low-N substrate —** | | | | | | |
| (*Sphagnum* peat) | | | | | | |
| low-N leaf | $0.96 \pm 0.03$ | $1.18 \pm 0.06$ | $53.3 \pm 2.4$ | $43.2 \pm 2.5$ | $32.5 \pm 2.0$ | $16.7 \pm 3.1$ |
| medium-N leaf | $1.39 \pm 0.07$ | $1.20 \pm 0.06$ | $35.5 \pm 0.9$ | $41.1 \pm 2.3$ | $21.1 \pm 0.9$ | $31.8 \pm 3.4$ |
| high-N leaf* | $2.14 \pm 0.11$ | $2.15 \pm 0.09$ | $25.5 \pm 1.7$ | $25.9 \pm 1.0$ | $25.7 \pm 3.4$ | $27.0 \pm 3.8$ |
| **Rhizomes - decomposed in substrates from their home site:** | | | | | | |
| Rhizome – high-N substrate* | $0.29 \pm 0.03$ | $1.18 \pm 0.31$ | $174 \pm 16$ | $47.6 \pm 10.5$ | $63.6 \pm 11.8$ | $-28.8 \pm 18.6$ |
| Rhizome – medium-N substrate* | $0.15 \pm 0.01$ | $0.81 \pm 0.11$ | $332 \pm 28$ | $66.3 \pm 6.0$ | $66.9 \pm 6.2$ | $-63.4 \pm 20.5$ |
| Rhizome – low N-substrate* | $1.37 \pm 0.02$ | $0.30 \pm 0.01$ | $37.1 \pm 0.8$ | $178 \pm 5$ | $33.3 \pm 1.5$ | $86.1 \pm 0.1$ |

\* Decomposition in home soil.

# 3 Results and discussion

## 3.1 Bulk C and N losses from litter

*P. australis* leaves and rhizomes were anaerobically decomposed in three anoxic, waterlogged soil substrates, i.e. two peat substrates and a detritus mud. The detritus mud, sampled from a rewetted fen, had a carbon:nitrogen (C/N) ratio of 14.4 and 72

5  mg/L $NH_4^+$-N in the porewater. A sedge-brown moss peat substrate had a C/N of 23.1 and 1.4 mg/L $NH_4^+$-N, and a *Sphagnum* peat a C/N of 38.6 and about 1.0 mg/L $NH_4^+$-N. Further site information is provided in Table S1. For convenience, we will term the three decomposition environments "high-N substrate", "medium-N substrate", and "low-N substrate". Senescent *P.*





*australis* leaves from these sites had initial C/N ratios of 35.5, 53.3, and 25.5. These three leaf litters will be termed "medium-N leaf", "low-N leaf", and "high-N leaf". *P. australis* rhizome litters from these sites had C/N ratios of 174, 332, and 37.1, respectively.

After 75 days of anaerobic decomposition, leaf litter C losses ranged from 21% to 45% (Table 1). In each soil substrate, the medium-N leaves showed the lowest litter C loss, which ranged from 21 - 34%. For low-N and high-N leaves, decomposed in the medium-N and high-N substrates, C losses were very similar ranging from 40 - 45%. In the low-N substrate, i.e. the *Sphagnum* peat, each leaf litter type showed the lowest C loss with values ranging from 21 - 33%.

The initial litter C/N ratio is often used as a measure for litter quality (Güsewell and Koerselman, 2002; McJannet et al., 1995; Rejmánková and Houdková, 2006; Aerts, 1997; Coûteaux et al., 1995). It is assumed that a higher initial N content with respect to C content leads to more N being available for microbial decomposers during enzymatic litter breakdown what generates favourable decomposition conditions. In our study, this trend is not detected. The highest litter C losses were observed for leaf litter lowest in initial N content. This suggests that leaf litter decomposition in this experiment was not N limited in terms of initial litter N content, probably due to the prevailing anoxic conditions. Anaerobic decomposers gain less energy per mole catabolized organic matter, thus a higher proportion of the depolymerized organic matter must be oxidized and less C remains for biomass growth. CUE values for anaerobic decomposers range from 5 - 10% and this low value also reduces the required amount of N the litter must provide (Reddy and DeLaune, 2008; Manzoni and Porporato, 2009).

In line with these considerations, total leaf litter N losses in this experiment were positively correlated with the initial litter N content ($R^2 = 0.51$, $p < 0.001$). In the medium-N soil, for example, net N losses accounted for 4, 20, and 33% for litter of an initial N content of 0.96, 1.39, and 2.14%. With respect to stoichiometric considerations, it can be concluded that, during litter breakdown, excess N was depolymerized from litter of higher initial N content and N mineralization gained in significance.

Another N flux detected in this study is the immobilization of external N from the soil porewater. Net N immobilization is observed for two rhizome litters. For these samples, the absolute amount of N within the decomposed tissue exceeded the N amount in the initial, undecomposed litter, denoted by negative N loss values in Table 1. Rhizomes decomposed in the N-poor *Sphagnum* peat, on the other hand, had an initially high N content, but showed the highest net N loss of 86.1%.

For leaf litter samples, the impact of gross immobilization of N from the environment into the litter can not unequivocally be determined from C and N loss data. Yet, a site-effect on N dynamics is apparent because for each leaf litter type the net N losses increased with decreasing site N availability. This is most notable when litter decomposition in the medium-N and high-N substrate (sedge-brown moss peat vs. detritus mud) is directly compared. The high-N leaves, for example, showed a similar C loss in both substrates. Yet, net N losses differed, accounting for 33% N after decomposition in the medium-N substrate, but only 24% N in the high-N substrates. The litter N content of this sample increased from 2.14% to 2.83% N (wt%) for decomposition in the high-N substrate, but only to 2.66% in the medium-N substrate. In the low-N substrate, on the other hand, net N losses from litter were comparably high and even exceeded C losses for the medium-N leaf litter. The N content of this sample dropped from initially 1.39 to 1.20% after 75 decomposition days, a finding that cannot reasonably be explained by variations in CUE or NUE. A net N loss that exceeds the net C loss is an indicator of preferential protein depolymerization.





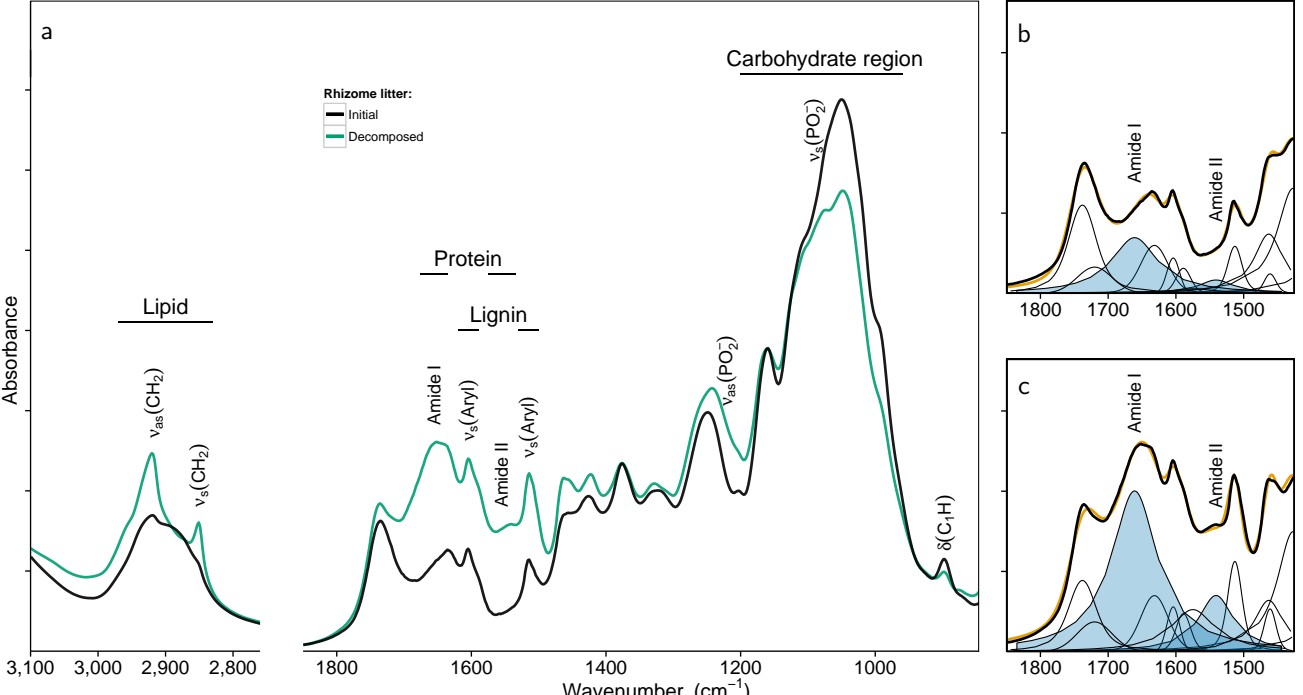

**Figure 2.** FTIR spectroscopic analysis of changes in rhizome litter chemical composition after decomposition in the medium-N soil. **(a)**: FTIR absorption spectra. Peak fitting of the FTIR spectra of the initial and decomposed rhizome litter. **(b & c)**: Plotted bands in blue denote the amide I and amide II bands of proteins.

## 3.2 Molecular characterization of initial and decomposed litter by FTIR spectroscopy

Changing litter C/N ratios are one indicator that the chemical composition of litter changed during decomposition. FTIR spectroscopy was used to gain a deeper insight into these changes. FTIR absorption bands can be assigned to molecular subunits present in the litter which in some cases are specific for single biopolymers like lignin, carbohydrates, lipids, proteins

5 or nucleic acids (Naumann, 2006; Pandey and Pitman, 2003; Whelan et al., 2014). Intensity changes of such bands due to decomposition allow a semiquantitative analysis of the enrichment or depletion of the specific biopolymer. Figure 2a shows the spectra of initial and decomposed rhizomes from the medium-N site. High C losses of 67% and the increase of bulk N from initially 0.29 to 1.18% are associated with substantial shifts in chemical composition. The preferential loss of cellulose and hemicellulose is revealed by decreasing absorption intensities in the region 1200-900 cm$^{-1}$. Lignin, on the other hand,

10 relatively accumulated, best seen by the increase of the aromatic sceletal vibration $\nu_s$(Aryl) at 1515 cm$^{-1}$. These findings are in agreement with other decomposition studies in that the mass loss in the early stage of decompositon is mainly a result of carbohydrate decomposition while more decay resistant lignin accumulates (Berg and McClaugherty, 2003).

Lignin and carbohydrate signals in FTIR spectra predominantly originate from plant tissue. Microbial biomass does contain

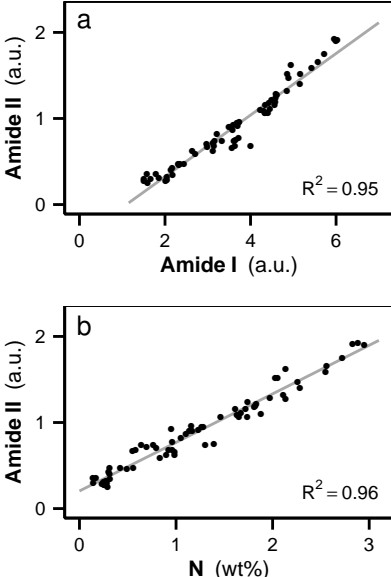

**Figure 3. (a)**: Correlation between amide I and amide II over all undecomposed and decomposed litters, and **(b)**: the correlation between bulk litter N content and amide II. a.u. = arbitrary units.

10-20% carbohydrates, but with respect to the high amount of carbohydrates in plant tissue, the contribution of microbial carbohydrates in the spectra of decomposed litter can be neglected. The same is true for lignin which is solely found in plant tissue.

For two other classes of biopolymers, this is different. Lipids and proteins are abundant in both, plant tissue and microbial

biomass. Lipids give rise to two sharp bands in the spectral region 3000-2800 $cm^{-1}$ (the symmetric and antisymmetric $CH_2$ stretches). While lipids are a prominent component of plant tissue (waxes), their bands were nearly absent in undecomposed rhizome spectra (Figure 2a). The substantial increase of these bands due to decomposition is unlikely caused by selective preservation of plant lipids but implies the accumulation of microbial biomass which has a lipid content of 10-15% dry weight (Naumann, 2006). This conclusion is in line with the prevailing perception in soil science that the accumulation of aliphatics

in soil organic matter is not caused by selective preservation of plant lipids, but due to microbial cell constituents making up a large to dominant fraction of aliphatic soil organic matter (Marín-Spiotta et al., 2014).

Similarly, proteins are present in plant tissue, but also account for 40-60% dry weight in microbial biomass (Naumann, 2006). The amide I band at 1661 $cm^{-1}$ and the amide II band at 1541 $cm^{-1}$ are the two major FTIR signals of proteins. These bands are present in the spectra of undecomposed rhizomes, but their strong increase due to decomposition indicate a substantial

protein enrichment within the litter, another indicator of the accumulation of microbial biomass and an expected pattern as bulk N data revealed net N immobilization having occured in rhizome litter decomposed in the medium-N substrate.



The amide I and amide II bands of proteins lie in a spectral region of high band overlap and are rather broad compared to adjacent signals. To more closely analyze protein band changes, we performed a peak fitting routine (Figure 2b & c). The fitted amide I and amide II band intensities were highly correlated with each other over all initial and decomposed litters ($R^2 = 0.95$, $p < 0.001$, Figure 3a). This corroborates the peak fitting since those two bands arise from the same molecular subunit and should

change synchronously. More importantly, we found a highly linear correlation between the amidic bands and the bulk litter N content for all initial and decomposed samples ($R^2 = 0.96$, $p < 0.001$, Figure 3b). This correlation suggests that variations in initial N content of undecomposed litters, as well as N changes due to decomposition, parallel and probably are governed by changes in litter protein content. This result resembles a 4 year subaqueous litter decomposition study in which litter protein content, quantified as total hydrolyzable amino acids (THAA), made up 52% of the bulk litter N content over the complete

course of decomposition, despite high mass losses and high changes in litter N content (Tremblay and Benner, 2006).
Infrared spectroscopy consequently confirms that changes of bulk litter N content parallel changes in protein content, that are a result of enzymatic protein depolymerization and microbial protein resynthesis, a finding that rationalizes the discussion of bulk litter N changes directly in terms of litter protein dynamics. It however does not directly provide new insights into gross N transformations during litter decomposition. Similar to protein quantification as THAA, amide bands in FTIR spectra are

not capable of distinguishing plant and microbial proteins within the litter but only detect bulk changes (Tremblay and Benner, 2006).

### 3.3   Quantification of microbial N in litter from FTIR

Here we wish to propose a novel approach to distinguish plant N and microbial N in decomposed litter. Using the concept of microbial homeostasis, i.e. the constant biopolymeric composition of microbial cells, the amount of microbial N in litter should increase proportionally to all other microbial cell constituents (Mooshammer et al., 2014b). The amount of microbial N

in a sample can thus indirectly be estimated from any other microbial biomass constituent that is absent in plant tissue. Compounds like muramic acid or the D-enantiomers of amino acids have previously been used for that purpose as these are absent in plant tissue but found in bacterial cells (Tremblay and Benner, 2006). Other markers to quantify living microbial biomass are phospholipid fatty acids (PLFA) or the amount of DNA in a sample. DNA accounts for 2-4% dry weight in microbial cells

(Naumann, 2006) but is only minorly present in senescent plant tissue. Either directly extracted from a sample or measured from $^{18}$O incorporation after adding isotopically labeled water, DNA has been used as a predictor of microbial C in previous studies (Marstorp et al., 2000; Spohn et al., 2016).
DNA can be analyzed using FTIR, as it gives rise to two prominent signals which are the antisymmetric and symmetric phosphodiester band of the DNA backbone, $\nu_{as}(PO_2^-)$ at 1220 cm$^{-1}$ and $\nu_s(PO_2^-)$ at 1090 cm$^{-1}$ (Whelan et al., 2014). FTIR

bands of DNA have not previously been reported in the spectra of decomposed litter because the amount of DNA is very small in such samples and more intense absorption bands of other biomolecules overlap, dominantly carbohydrate signals (see Figure 2a). Yet, DNA bands are detectable in the spectra of isolated microbial cells and have extensively been studied (Naumann, 2006). Notably the asymmetric phosphodiester band at 1220 cm$^{-1}$ has been shown to be a semiquantitative marker of cellular DNA content (Zucchiatti et al., 2016).

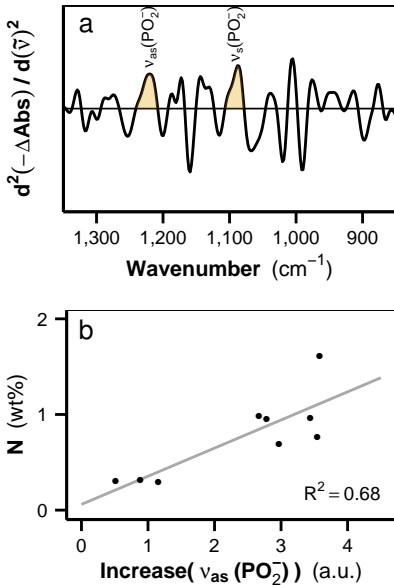

**Figure 4.** **(a)**: Second derivative difference spectra show positive bands at 1220 and 1090 cm$^{-1}$, identified as phosphodiester stretching vibrations of DNA. The band height of the asymmetric phosphodiester band at 1220 cm$^{-1}$ is used as marker for increases of microbial biomass content in litter. **(b)**: Correlation between $\nu_{as}(PO_2^-)$-increase and N content in decomposed rhizome litters. a.u. = arbitrary units.

In litter FTIR spectra, the two phosphodiester bands of DNA could be resolved through FTIR difference spectroscopy. FTIR difference spectroscopy is a method to overcome the problem that small changes in the molecular structure of a complex tissue are difficult to study as the corresponding spectral changes are very small compared to the signals of the unchanged residual organic matter (Moss et al., 2000). Yet, a subtraction of two FTIR spectra (unmodified and modified tissue) leads to a FTIR

difference spectrum that selectively shows spectral changes. The litter FTIR difference spectra (decomposed minus undecomposed litter) show negative bands for biomolecules that have preferentially been lost and positive bands for biomolecules that relatively accumulated or newly emerged (Figure S1). In particular, a positive band at 1220 cm$^{-1}$ was detected which we identified as the antisymmetric phosphodiester band of DNA. The second DNA band at 1090 cm$^{-1}$ appeared as a shoulder. In order to remove baseline effects and to increase the resolution of the DNA bands, we formed the second derivatives of the FTIR

difference spectra (Susi and Byler, 1983), wherein the height of the band at 1220 cm$^{-1}$ was used as a semiquantitative marker for the content of newly formed DNA in the samples (Figure 4a & Figure S1).

Ultimately, the determined 1220 cm$^{-1}$ band height, as a measure for microbial DNA, needed to be calibrated against the corresponding amount of newly formed microbial N. When using other microbial biomass markers like PLFA or muramic acid, such a calibration is performed on cultured microbial cells (Tremblay and Benner, 2006; Rejmánková and Houdková, 2006). FTIR

spectra of isolated microbial cells were unavailable and, furthermore, it is likely that such an approach would lead to unsatis-





fying results due to the strong plant matrix in litter which most likely effects the DNA band height in a spectroscopic method like FTIR, why an in-situ calibration was needed. In this study we used the decomposed rhizome litter spectra as "calibration standards" assuming that these contained only neglectable amounts of remaining plant N. This assumption possibly leads to an overestimation of microbial N in the later decribed results as some plant-bound N might be remaining in the decomposed

rhizomes. In former studies, it was regularly assumed that the relative plant N content within litter remains constant during decomposition. Here, bulk N content in one rhizome litter increased from 0.15% to 0.81%, so we equally could have assumed a microbial N content of 0.66% and 0.15% N being plant-bound. Our method would in this case overestimate microbial N in leaf litter by about 18%. However, we tested the effect of a correspondingly lower slope of the calibration curve during data evaluation and all here presented trends remained unaffected. Furthermore, rhizome decomposition in the N-poor soil resulted

in a decreasing litter N from initially 1.37% to 0.30%, what shows that a preferential loss of plant-N from the litter is a possible process. Therefore assuming that all N in decomposed rhizomes was microbial N, we calculated microbial N in decomposed leaf litter as "$N_{microbial} = 0.3145 \cdot \text{Increase}(\nu_{as}(PO_2{}^-))$" (Figure 4b).

### 3.4 Microbial N in decomposed litter

Applying the calibration on leaf litter, the determined microbial N in decomposed leaves ranged from 0.16 to 0.72% litter dry

mass or 14 to 42% of the total litter N (Table 2). The disentangling of plant and microbial N, presented graphically in Figure 5, shows that the relative increase in N content due to decomposition is indeed dominantly caused by an accumulation of microbial N within the litter tissue. The relative microbial N contents between samples are, however, not directly representative for the total amount of microbial N, because each litter sample has a distinct remaining mass. Microbial N content was lowest after decomposition in the low-N soil for each litter type (rightmost bar in Figure 5a, b and c). Yet, litter C losses were also lowest

for these samples, so a low percentual microbial N content could represent a comparibly high total N amount as the overall remaining litter mass is very high. On the other hand, stoichiometric considerations predict lower amounts of microbial biomass and thus microbial N for litters with lower C loss as the total decomposition process is less advanced.

In order to place the microbial N content of different samples into a relationship with one another and to take variations in C loss into account, we assumed microbial homeostasis and a microbial C/N ratio of 5 (Mouginot et al., 2014). The thereby

determinable total amount of newly formed microbial C was set in relation to the litter C loss, what leads to an estimate of the CUE (Manzoni, 2017). The CUE was determined as:

$$CUE = \frac{C_{microbial}}{C_{depolymerized}} = \frac{5 \cdot N_{microbial}}{C_{loss} + 5 \cdot N_{microbial}}$$

The determined CUE values ranged from 6.6 to 10.0% and hence are within the expected range of values for anaerobic decomposition (Reddy and DeLaune, 2008) (Table 2). The CUEs were slightly affected by the leaf litter type. The medium N

leaf litter which in each soil showed the lowest C loss also showed lowest CUE values. Consequently, the C loss as well as the CUE indicate, that this leaf litter type is less well decomposable compared to other leaves (Sinsabaugh et al., 2013; Manzoni et al., 2012; Takriti et al., 2018).

Beside this substrate quality trend, no significant effect of C loss or decomposition site on CUE was observed (both p>0.05).

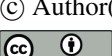



**Table 2.** Selected C and N data of leaf litter, of its microbial N content, microbial CUEs and nNUEs, and parameters of the plant biomass fraction of litter. The litter encompasses the remaining plant biomass and the newly formed microbial biomass.

| Litter Type and Decomp. Environment | Litter[a] | | | | Microbial biomass | | | Plant biomass | | | | |
|---|---|---|---|---|---|---|---|---|---|---|---|---|
| | $N_{litter}$[a] | $C_{loss}$ | $N_{loss}$ | $(C/N)_{litter}$ | $N_{microbial}$[b] | CUE[c] | nNUE[d] | $N_{plant}$[e] | $C_{depoly.}$[f] | $N_{depoly.}$[g] | $\alpha$[h] | $(C/N)_{plant}$[i] |
| | (wt%) | (%) | (%) | (atom) | (wt%) | (%) | (%) | (wt%) | (%) | (%) | (ratio) | (atom) |
| —low-N leaf litter — | | | | | | | | | | | | |
| initial | 0.96 ± 0.03 | 0 | 0 | 53.3 ± 2.4 | 0 | – | – | 0.96 ± 0.03 | 0 | 0 | – | 53.3 ± 2.4 |
| high-N substrate | 1.71 ± 0.03 | 44.0 ± 1.7 | 1.4 ± 3.6 | 30.2 ± 0.2 | 0.67 ± 0.02 | 8.77 ± 0.61 | 96.9 ± 9.1 | 1.04 ± 0.01 | 48.3 ± 1.5 | 40.0 ± 2.3 | 0.83 ± 0.02 | 45.9 ± 0.4 |
| medium-N substrate | 1.67 ± 0.05 | 45.0 ± 2.4 | 3.8 ± 4.3 | 30.4 ± 0.8 | 0.72 ± 0.04 | 9.21 ± 1.13 | 92.0 ± 8.4 | 0.95 ± 0.08 | 49.6 ± 2.1 | 45.3 ± 4.3 | 0.92 ± 0.09 | 49.3 ± 3.6 |
| low-N substrate | 1.18 ± 0.07 | 32.5 ± 2.0 | 16.7 ± 3.1 | 43.2 ± 2.5 | 0.46 ± 0.01 | 9.99 ± 0.92 | 66.5 ± 4.6 | 0.71 ± 0.06 | 36.1 ± 1.9 | 49.5 ± 3.3 | 1.38 ± 0.16 | 67.7 ± 6.2 |
| — medium-N leaf litter — | | | | | | | | | | | | |
| initial | 1.39 ± 0.07 | 0 | 0 | 35.5 ± 0.9 | 0 | – | – | 1.39 ± 0.06 | 0 | 0 | – | 35.5 ± 0.9 |
| high-N substrate | 1.97 ± 0.13 | 33.9 ± 1.5 | 11.6 ± 4.5 | 26.5 ± 1.3 | 0.35 ± 0.03 | 7.02 ± 0.78 | 58.6 ± 9.9 | 1.63 ± 0.11 | 36.5 ± 1.4 | 27.1 ± 4.1 | 0.74 ± 0.11 | 31.0 ± 1.8 |
| medium-N substrate | 1.84 ± 0.03 | 38.7 ± 0.5 | 20.3 ± 2.0 | 27.2 ± 0.5 | 0.41 ± 0.00 | 6.93 ± 0.19 | 46.5 ± 2.9 | 1.44 ± 0.03 | 41.5 ± 0.5 | 37.8 ± 1.8 | 0.91 ± 0.03 | 33.3 ± 0.7 |
| low-N substrate | 1.20 ± 0.06 | 21.1 ± 0.9 | 31.8 ± 3.4 | 41.1 ± 2.3 | 0.16 ± 0.06 | 6.58 ± 1.94 | 22.0 ± 5.5 | 1.04 ± 0.09 | 22.6 ± 1.4 | 41.0 ± 5.2 | 1.82 ± 0.21 | 46.8 ± 3.7 |
| — high-N leaf litter — | | | | | | | | | | | | |
| initial | 2.14 ± 0.11 | 0 | 0 | 25.5 ± 1.7 | 0 | – | – | 2.14 ± 0.11 | 0 | 0 | – | 25.5 ± 1.7 |
| high-N substrate | 2.83 ± 0.09 | 40.2 ± 1.9 | 23.5 ± 1.6 | 19.9 ± 0.5 | 0.55 ± 0.06 | 7.81 ± 0.20 | 38.7 ± 1.9 | 2.28 ± 0.05 | 43.6 ± 2.1 | 38.3 ± 2.2 | 0.88 ± 0.02 | 23.2 ± 0.4 |
| medium-N substrate | 2.66 ± 0.16 | 43.0 ± 2.2 | 33.3 ± 0.7 | 21.7 ± 1.1 | 0.63 ± 0.05 | 7.80 ± 1.09 | 32.1 ± 2.1 | 2.04 ± 0.18 | 46.6 ± 1.9 | 49.0 ± 2.3 | 1.06 ± 0.09 | 26.7 ± 2.0 |
| low-N substrate | 2.15 ± 0.09 | 25.7 ± 3.4 | 27.0 ± 3.8 | 25.9 ± 1.0 | 0.36 ± 0.03 | 9.97 ± 1.13 | 31.5 ± 4.3 | 1.79 ± 0.06 | 28.5 ± 3.4 | 39.3 ± 3.4 | 1.39 ± 0.11 | 30.0 ± 1.0 |

[a] Decomposed litter contains plant and microbial N.

[b] $N_{microbial}$: N content in decomposed litter belonging to microorganisms (in wt% litter dry mass).

[c] CUE: Carbon use-efficiency.

[d] nNUE: Net nitrogen use-efficiency.

[e] $N_{plant}$: N content in decomposed litter belonging to plant OM (in wt% litter dry mass).

[f] $C_{depoly.}$: Fraction of initially present plant C which has been depolymerized.

[g] $N_{depoly.}$: Analogously to $C_{depoly.}$

[h] ($\alpha$): Coefficient of preferential protein decomposition ($N_{depoly}/C_{depoly}$).

[i] $(C/N)_{plant}$: Carbon-to-nitrogen ratio of the plant OM fraction within the litter.





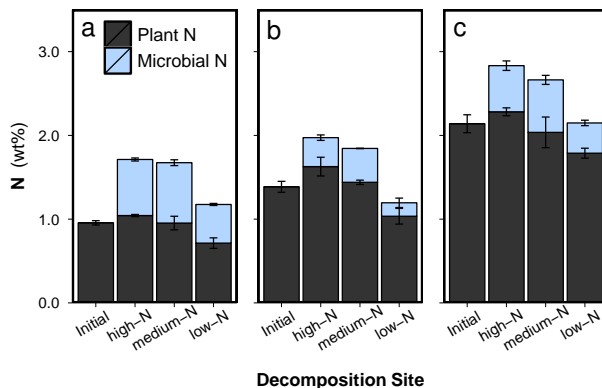

**Figure 5.** Microbial N content and plant N content in leaf litter after 75 days of anaerobic decomposition in three substrates of varying N content (decomposition site). **(a)**: low-N leaf litter, **(b)**: medium-N leaf litter, and **(c)**: high-N leaf litter. Error bars represent standard deviations of three replicate litterbags.

Each litter sample was decomposed at a rather constant CUE, independently of the biogeochemical variations of the soil in which decomposition took place. This finding is in line with the results of other studies like Spohn et al. (2016) in which incorporation of $^{18}$O from labeled water into microbial DNA was compared to $CO_2$ formation and in which it was analogously found that the CUE can remain constant despite variations in substrate C/N ratio or decomposition rate. In our study, in which

a higher accumulation of total N in decomposing litter was observed in soils with a higher external nutrient availability, the constant CUEs indicate that these variations in N dynamics are not due to variations in microbial N accumulation.

Microbial N contents furthermore allow the calculation of the net nitrogen use-efficiency (nNUE). Analogously to the CUE, the nNUE compares the amount of newly formed microbial N to the amount of depolymerized N.

$$nNUE = \frac{N_{microbial}}{N_{depolymerized}} = \frac{N_{microbial}}{N_{loss} + N_{microbial}}$$

Even though it is predicted by theory, that the NUE will dominantly depend on the initial litter N content, empirical evidence for that assumption was only very recently reported from isotope pool dilution experiments with labeled amino acids (Mooshammer et al., 2014a, 2012; Wanek et al., 2010). Pool dilution experiments allow the analysis of gross N fluxes like amino acid formation (i.e. gross protein depolymerization) or microbial amino acid consumption and confirm the interplay between litter N content, the proportion of N used for growth (NUE), and N mineralization. In this study, we use the parameter nNUE as our

litter-centered analytical approach does not allow to distinguish wether the source of the microbial N is depolymerized plant protein or inorganic N from the environment. nNUE values ranged from 97% (as no net N loss occurred in some litters) to 22% (Table 2). In each soil, the nNUE decreased with increasing initial litter N content. Litters with initial N content of 0.96, 1.39 and 2.14%, for example, revealed nNUEs of 92, 47 and 32% when decomposed in the medium-N substrate, respectively. As a second trend, however, a decomposition site effect on nNUE is evident as for each leaf litter, the nNUE decreased with

decreasing external N availability. The medium-N leaf litter, for example, showed an nNUE of 59, 47 and 22% for decomposi-





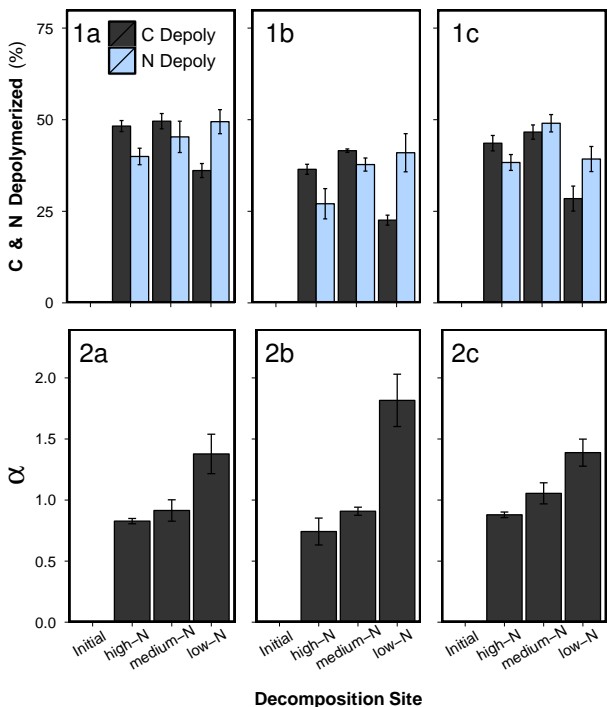

**Figure 6. (1)**: Depolymerized plant C and depolymerized plant N after 75 days of anaerobic litter decomposition, and **(2)**: the coefficient of preferential protein depolymerization $\alpha$ (= $N_{depoly}$/$C_{depoly}$). **(a)**: low-N leaf litter, **(b)**: medium-N leaf litter, and **(c)**: high-N leaf litter. Error bars represent standard deviations of three replicate litterbags.

tion in the high-, medium- and low-N substrate. In part, a decreasing immobilization of N from the porewater might contribute to this pattern. However, the site-dependent nature of the nNUE goes along with constant CUEs, what suggests, that microbial N accumulated according to the theoretical predictions (Manzoni et al., 2008). The single explanation for a lower nNUE in a more nutrient-poor environment, which goes along with normal microbial N accumulation rates and a lower N accumulation
5  in the overall litter tissue, is a site specifically high rate of plant protein depolymerization.

### 3.5 Depolymerized C and N & preferential protein depolymerization

We mentioned in the introduction that the critical difference between C cycling and N cycling in decomposing litter lies in the fact that the C loss is an acceptable measure for C-compound depolymerization what does not apply to N loss for N depolymerization. In our study, estimated amounts of microbial C and microbial N allow a comparison of loss and depolymerization.
10  The low-N leaf litter, decomposed in the high-N soil, for example, lost 44% C and 1.4% N during decomposition (Table 1). The bulk N content of this litter increased from initially 0.96% to 1.71%. The decomposed litter had a bulk C content of 44.4%, of which 3.4% is now identified as microbial C. By summing up total amounts of microbial C and lost C, it follows that 48%



of the initially present litter C has already been depolymerized at this stage of decomposition, a value slightly higher than the C loss of 44% (Table 2). Due to the low CUE of 8.77%, there is only a minor difference between depolymerized C and C loss. The bulk litter N content increased from 0.96% to 1.71%. This corresponds to almost no N being lost from the litter (1.4%). Yet, microbial N in this sample was determined as 0.67%, what implies that the plant N content in the tissue accounts for

1.04%, a value only minorly higher than that in the undecomposed litter. Using a plant N of 1.04%, it follows that 40% of the initially present N has been depolymerized (Table 2). At an nNUE of 96.9%, nearly all depolymerized N was used for microbial growth (on a net basis) and thus remained within the litter.

Percentual amounts of depolymerized C and N of all leaf litters are presented in Table 2 and graphically in Figure 6.1a-c. The patterns of depolymerized C are similar to the C loss patterns. N-compound depolymerization, however, proceeded at a

different rate than C-compound depolymerization. In particular, the percentual amount of depolymerized N after 75 days of decomposition increased along the decomposition site nutrient status from high-N substrate to low-N substrate for each litter type.

In order to characterize the variation between C depolymerization and N depolymerization, we use the coefficient of preferential protein depolymerization ($\alpha$). This parameter is the ratio "depolymerized N / depolymerized C" and is adopted from recent

stoichiometric models which introduce this term as the coefficient of preferential N uptake (Manzoni, 2017; Manzoni et al., 2010). Our study for the first time provides empirical data for $\alpha$. $\alpha$ ranged from 0.74 to 1.82, did not depend on the initial litter N content ($R^2 = 0.004$, $p > 0.05$), but appeared to be a decomposition site property, which correlated negatively with the N content of the soil in which the litter decomposed ($R^2 = 0.76$, $p < 0.001$) (Figure 6.2a-c & Table 2). Preferential protein depolymerization as a decomposition site property is here shown as one aspect on how the environment can affect N dynamics

in decomposing litter: microbial decomposers apparently adjust their protein decomposition activity in response to the environmental N availability. Even though not reported in long term litter decomposition studies, such adaption mechanisms are generally known. Microbes are capable of varying their enzyme production to specifically target resources in shortest supply (Schimel and Weintraub, 2003; Liang et al., 2017; Waring, 2013; Rejmánková and Sirová, 2007). Our data demonstate, that as a consequence of the intensification of microbial protein depolymerization activity in the N-poor environments (*Sphagnum*

peat), the remaining still unprocessed plant organic matter in litter becomes gradually N depleted, what leads to the observed site dependent variations in the bulk N dynamics of the overall litter (see C/N$_{plant}$-data in Table 2). The possibility of variations in protein depolymerization in relation to environmental N availability have, however, received very little attention as a possible explanation for site-dependent variations in litter N dynamics, even though higher N accumulation in litter as a function of increasing external N availability is regularly observed in field studies (Hobbie and Gough, 2004; Manzoni et al.,

2008; Rejmánková and Houdková, 2006; Ågren et al., 2013; Moore et al., 2006).

In the high-N soil we found $\alpha$ values lower than 1, what indicates that plant proteins are more slowly depolymerized than plant C compounds. It is indeed known that high concentrations of $NH_4^+$ inhibit protein depolymerization activity (Geisseler et al., 2010). The reduced plant protein depolymerization activity at 72 mg/L $NH_4^+$-N in the porewater of the high-N soil is a likely mechanism causing the observed higher N accumulation in decomposing litter.

Litter N dynamics in the low-N *Sphagnum* peat are distinct. Compared to decomposition in other soils, each litter showed




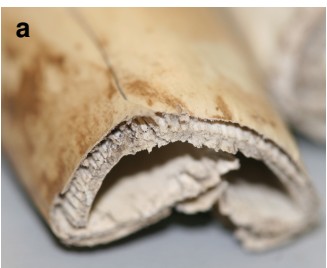

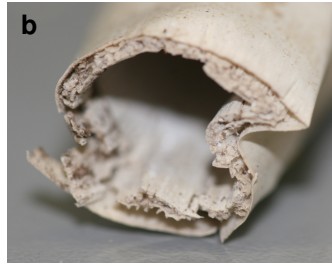

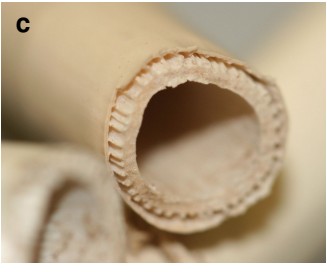

**Figure 7.** *P. australis* rhizomes after 75 days of anaerobic decomposition in the three peat substrates. **(a)**: high-N substrate (detritus mud), **(b)**: medium-N substrate (sedge-brown moss peat), and **(c)**: low-N substrate (*Sphagnum* peat).

high values of depolymerized protein, low values of depolymerized C compounds, and thus high $\alpha$-values (Figure 6). High $\alpha$-values between 1.38 and 1.82 in an environment which provides very little external N is rational and not exceptionally high in relation to $\alpha$-values of 1.3 assumed in some stoichiometric models (Manzoni, 2017). Yet, the increased protein depolymerization activity is accompanied by very low nNUEs (Table 2). Only a minor fraction of the depolymerized N is actually used

5 for microbial growth. This interaction partly leads to net N losses from litter that exceed C losses and result in an overall increasing litter C/N ratio, what is an uncommon finding. Why should microbes in *Sphagnum* peat preferentially depolymerize N from vascular plant tissue only to excrete it to the environment? A possible explanation for this pattern is not the low external nutrient availability, but distinct porewater properties of *Sphagnum* peatlands. It is well known, that litter decomposition in *Sphagnum*-dominated peatlands is low, what is dominantly contributed to the nutrient-poor, anoxic and acidic conditions.

10 *Sphagnum* mosses, however, additionally release sphagnan into the porewater, which is a pectic-like cell wall polysaccharide of the moss (Hájek et al., 2011). Sphagnan has been shown as being capable of interacting with free amino groups. An observed inhibiting effect on litter decomposition by sphagnan is accounted for an inactivation of plant tissue-depolymerizing





exoenzymes, which are composed of proteins (Hájek et al., 2011). Building on these findings, our study suggest a different mechanism of inhibition by sphagnan, which is an interaction not with the decomposing exoenzymes, but with the depolymerized amino acids, which once complexed to sphagnan cannot further be used by microbial decomposers. This would cause a competition-type environment for microbial decomposers with respect to N-accquisition because part of the depolymerized

amino acids become inaccessible. An excess protein depolymerization rate might be an adaption to overcome N-limitation. The by Hájek et al. (2011) reported accumulation of dissolved organic nitrogen, apparently stable to further decomposition, would in this case originate from amino acids complexed to sphagnan and not from complexed exoenzymes as proposed by the authors. Preferential protein depolymerization as an adaptive mechanisms to amino acid complexation by sphagnan would furthermore explain the results of other decomposition studies in *Sphagnum* peat, which similarly found unexpectedly high

total N losses from vascular litter during decomposition, which however were not discussed with respect to sphagnan activity or preferential protein depolymerization (Scheffer et al., 2001; Verhoeven et al., 1990).

It is important to notice, that preferential protein depolymerization leads to a gradual N depletion of the remaining undecomposed litter, denoted by increasing plant tissue C/N ratios in Table 2. As N is an essential element for microbial decomposers, this N depletion can be regarded as a gradual decrease of litter decomposability over the course of decomposition. Even the

decomposition of N-rich litter will become N limited and significantly slowed down long before all plant C has been depolymerized. This process might be an explanation for the observed low decomposition of the rhizome litter in this substrate. While initially rather N rich (1.37% N), this litter lost 89% of its N during decomposition, while C losses only accounted for 33%. In line with the low C loss, this sample was physically rather intact compared to rhizomes decomposed in the two other substrates (Figure 7). As this litter reached a final C/N ratio of 178 at the end of the experiment, it can be assumed, that even at longer

decomposition times, a further decomposition of this litter might be very unfavourable due to the site-effected N depletion and a resulting N limitation. This site-induced reduction of litter decomposability might be an important mechanisms of vascular plant litter preservation and C storage in *Sphagnum* peatlands.

## 4 Conclusions

In wetland ecosystems, the disentangling of gross N transformations is central for the assessment of C cycling, N cycling,

and biogeochemical responses to increased external nutrient inputs, especially of N (Van Groenigen et al., 2015). Here, we have presented a new methodical approach which enables the disentangling of plant and microbial N in decomposed litter by using DNA signals in FTIR spectra as a marker for microbial N. This approach allows to quantify how much microbial N has formed and how much plant N is remaining in litter at a certain stage of decomposition.We have demonstrated that substrate-dependent, i.e. decomposition site dependent, variations in litter N accumulation were not caused by a variations in

the amount of existing microbial N in litter at a certain stage of decomposition, but instead by variations in the remaining amount of still unprocessed plant-N. This indicates a decrease of gross protein depolymerization when litter decomposes in a nutrient-rich environment, and suggests that microorganims preferentially use inorganic N from porewater instead of organic N from litter when those two pools compete as consumable N sources. The influence of the decomposition site and its nutrient
status on gross protein depolymerization in decomposing vascular litter has not been detectable in former studies. Instead, net N mineralization/immobilization patterns in decomposing litter are often perceived as being predetermined solely by the initial litter C/N ratio and the rate of microbial N accumulation.

For litter decomposition in *Sphagnum* peat, we found high bulk N losses from litter due to preferential protein depolymeriza-
tion. Although not directly adressed in this study, we assume that sphagnan, a compound released from *Sphagnum* mosses, is responsible for this effect. Sphagnan is known to bind free amino groups, so that a fraction of the depolymerized amino acids might become inaccessible for microbial decomposers. We suggest that preferential protein depolymerization might be an adaptive mechanism to compensate amino acid losses to sphagnan. Preferential protein depolymerization leads to a gradual N depletion of the unprocessed organic matter in litter, what can be considered as a gradual decrease of litter decomposability,
a potential mechanism for long-term preservation of vascular litter in *Sphagnum* peat.

*Data availability.*  Data of the decomposition experiment including raw data of FTIR absorbance and second derivative spectra and unused data of CuO lignin phenols in soil substrates and litter can be downloaded at https://doi.pangaea.de/10.1594/PANGAEA.902181.

*Author contributions.*  All authors participated in conceptual discussions, drafting and revising the manuscript. HR and DZ developed the research strategy and performed field sampling. HR performed laboratory work and wrote the manuscript. HR and JG performed data
analyses.

*Competing interests.*  No competing interests are present.

*Acknowledgements.*  We thank Jörg Gelbrecht for advice on sampling sites and experimental design and Michael Zauft for sampling per-mission in the protected area Töpchin Süd (EU Life+-Projekt "Kalkmoore Brandenburgs"). We furthermore appreciate comments by Robert Taube, Jos Verhoeven and Paul Bodelier on a previous version of the manuscript. This research was supported by the Deutsche Forschungs-
gemeinschaft (ZA 742/2-1).



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
