# Peer review of "Evidence for preferential protein depolymerization in wetland soils in response to external nitrogen availability provided by a novel FTIR routine"

_Biogeosciences, 2019_

## Referee Comment (RC1) · Tim Moore (Referee) · 26 Jun 2019

This is an interesting manuscript which explores the fate of N during the early stages of decomposition of plant litter under anaerobic conditions. In essence, the authors examine the depolymerisation of N in litters and substrates of varying N content and FTIR is adapted to provide identification of the chemical changes in the decomposing litter, with an estimate of the microbial uptake and depolymerisation components and stoichiometric relationships.

The manuscript is well structured and written and provides some insight into the de-polymerisation process, which has been proposed for several years, but which has

been difficult to analytically identify. I hasten to add that I have a very limited knowledge of the dark art of FTIR spectra analysis, so look to other reviewers to evaluate the veracity of the FTIR section of the manuscript. I have noted some grammatical/typographical and stylistic errors and suggestions, and place them in the pdfs, which are hopefully attached.

I provide the following more detailed comments and suggestions for consideration by page and line number:

1, 0 While I think that the preservation of vascular litter in Sphagnum peat is a useful product of the work, I think it has a broader impact, and most litter entering Sphagnum (and other) peats decays initially under aerobic conditions, rather than the anaerobic burial used in this experiment. Thus I would suggest a more generic title, emphasizing the more original approaches taken.

1, 2 What does 'relative' mean here? It could be N accumulation relative to N (implying a lowering of the C:N ratio) or it could be a larger N mass, relative to the initial litter. Please clarify.

4, 25 The experiment was conducted under anaerobic conditions, or at least litter placed in containers into which substrates had been added and presumably under saturated or waterlogged conditions. I think this is important, partly because of the conditions created (anoxic) and, as I note above, most peatland vascular litter does not decompose initially under anaerobic conditions. Thus, I think the experimental details of these containers and substrates/litter need to be better described. Also, were they incubated at 'room temperature'? Furthermore, are the results of this study likely to be repeated, quantitatively or qualitatively, if the experiment was to be repeated under aerobic conditions, which is probably the situation in many wetlands. Of course, one could argue that the initial aerobic decomposition is followed by anaerobic, as the litter becomes buried and goes beneath the water table.

Table 1. I was a bit confused by * decomposition in home soil. I would have thought

the 'home soil' would be high-N leaf with high-N substrate, medium with medium etc., but this is not the pattern observed. I wondered why.

7, 4 Litter bag experiments usually entail the early stages of decomposition, in this case 21 to 45% over 75 days. One wonders what the patterns may have been if the study allowed sampling earlier and later: in other words, are the processes identified here time-dependent in the decomposition path?

7, 8 Litter quality involves several attributes of the initial litter influencing decomposition rate, of which the C:N ratio is frequently cited. It was not borne out here, possibly because decomposition was under anaerobic conditions. Were there any other attributes of the litter which might explain this deviation, such as P content, lignin content etc.?

Table S1 While nitrate was essentially non-existent in the porewater from the three substrates, there was a major difference in NH4 and also DOC, the latter implying a large variation in dissolved organic nitrogen (DON), referred to p 18, l6. In Sphagnum peatlands, DON dominates the pore water, often forming 60-90% of the total dissolved nitrogen (TDN). It appears that TDN was not measured (allowing an estimate of DON) but could there be more consideration of DON in the understanding of the processes involved?

I found it a little bit confusing that C and N ratio was expressed atomically, whereas everything appears to be on a mass basis; while atomic units are common in stoichiometric studies, most decomposition studies use mass.

Sequence of reference citations seems to vary between alphabetical and chronological and the format used in the References is variable.

In case the Supplement does not load, oxycoccos is mis-spelt and it is Electrical conductivity.

Tim Moore

Please also note the supplement to this comment:
https://www.biogeosciences-discuss.net/bg-2019-176/bg-2019-176-RC1-
supplement.pdf

―――――――――――――――――――――

[Figure]

**Supplement:**

[revised manuscript text omitted]

---

## Author Comment (AC1) · 17 Jul 2019

**Discussion of comments by Tim Moore on bg-2019-176**

Hendrik Reuter, Julia Gensel, Marcus Elvert, and Dominik Zak

Reviewer comment: This is an interesting manuscript which explores the fate of N during the early stages of decomposition of plant litter under anaerobic conditions. In essence, the authors examine the depolymerisation of N in litters and substrates of varying N content and FTIR is adapted to provide identification of the chemical changes in the decomposing litter, with an estimate of the microbial uptake and depolymerisation components and stoichiometric relationships.

- 5 The manuscript is well structured and written and provides some insight into the depolymerisation process, which has been proposed for several years, but which has been difficult to analytically identify. I hasten to add that I have a very limited knowledge of the dark art of FTIR spectra analysis, so look to other reviewers to evaluate the veracity of the FTIR section of the manuscript. I have noted some grammatical/typographical and stylistic errors and suggestions, and place them in the pdfs, which are hopefully attached.
- 10 Authors reply: We thank you for your careful reading of the manuscript. We appreciate your comments on improvements of the writing, which are highly valuable for us as non-native speakers.

We also thank you for your detailed comments and suggestions, which we would like to discuss in detail. Please do not hesitate to post additional comments if you feel that some of our answers require further explanation or discussion.

As a preliminary outline: There are five main aspects which might be discussed in more detail in the manuscript. These are (1) the missing correlation between nutrient status of the wetland and N content of the corresponding undecomposed plant litter,

(2) the variations in litter quality/decomposability, i.e. why does the medium-N leaf litter decomposes more slowly than the other two leaf litters,

20 (3) the role of dissolved organic nitrogen,

30

- (4) the time-dependency of the reported process (preferential N decomposition) over the decomposition path, and
- (5) the repeatability of the results in an oxic decomposition experiment.

A reply to point (5) must include both, a discussion of the applicability of the FTIR method in an oxic decomposition study
and a discussion whether preferential N decomposition can be expected to be a relevant process in aerobic ecosystems. We suggest to change the "Conclusions" to "Conclusions and perspectives" in order to cover these aspects.

Point (3), the role of DON, will be adressed at selected points throughout the manuscript.

Point (1), (2), and (4) are interesting aspects, some of which had already been discussed by us during data analysis and manuscript writing. We believe that a discussion of these points might overload the main document and suggest to discuss these points solely here in the interactive discussion. As documents herein are citable, we suggest to add citations in the

manuscript at relevant positions that lead the interested reader to this file.

I provide the following more detailed comments and suggestions for consideration by page and line number:

1, 0 While I think that the preservation of vascular litter in Sphagnum peat is a useful product of the work, I think it has a broader impact, and most litter entering Sphagnum (and other) peats decays initially under aerobic conditions,

35 rather than the anaerobic burial used in this experiment. Thus I would suggest a more generic title, emphasizing the more original approaches taken.

We agree that the current title does not cover some aspects of the manuscript. We will change the title to:

- Unraveling preferential protein depolymerization from litter in response to external nitrogen availability in three anoxic wetland soils through a novel FTIR routine
- 40 1, 2 What does 'relative' mean here? It could be N accumulation relative to N (implying a lowering of the C:N ratio) or it could be a larger N mass, relative to the initial litter. Please clarify.

We thank you for this comment and will change the beginning of the abstract:

*Phragmites australis* litters were incubated in three saturated anoxic wetland soils of different nutrient status for 75 days and litter nitrogen (N) dynamics were analyzed by elemental analyses and infrared spectroscopy (FTIR). At the end of the incuba-

45 tion time, the N content in the remaining litter tissue had increased in most samples. Yet, the increase of N content was less pronounced when litters had been decomposed in a more N-poor wetland soil. FTIR was used to quantify the relative content of proteins...

4, 25 The experiment was conducted under anaerobic conditions, or at least litter placed in containers into which

- 50 substrates had been added and presumably under saturated or waterlogged conditions. I think this is important, partly because of the conditions created (anoxic) and, as I note above, most peatland vascular litter does not decompose initially under anaerobic conditions. Thus, I think the experimental details of these containers and substrates/litter need to be better described. Also, were they incubated at 'room temperature'? Furthermore, are the results of this study likely to be repeated, quantitatively or qualitatively, if the experiment was to be repeated under aerobic conditions, which is
- 55 probably the situation in many wetlands. Of course, one could argue that the initial aerobic decomposition is followed by anaerobic, as the litter becomes buried and goes beneath the water table. In order to visualize the details of the experiment more precisely we would like to present two photos. Figure 1a shows the

container filled with detritus mud (high-N substrate) after sampling and during the transport to the lab. Figure 1b shows the prepared litterbags with rhizome and leaf litter before incubation. The containers with sampled substrates were transported

60 to the basement of our institute in June 2013, and three days later the litterbags were placed in each container, close to the bottom. We will include Figure 1a to the "Materials and Methods" section in the revised manuscript and furthermore describe the experimental design in more detail.

The term "room temperature" will be changed to 21°C.

We agree that the placement of freshly fallen leaf litter into an anoxic environment does not mimic the common pattern of litter decomposition in wetlands. Naturally, leaf litter is initially decomposed in the actrotelm and only becomes buried after extensive aerobic decomposition. Only for the incubated rhizomes direct anaerobic decomposition can be expected. A rapid burial of leaf litter and its anaerobic decomposition might also happen in the rewetted fen where sedimentation rates are high.

70 Our study specifically aimed to compare the process of anaerobic decomposition in peatlands and to search for differences in the enzymatic breakdown of organic matter which might explain a part of the rapid conversion of organic matter to DOM,

65

Figure 1. (a) The container with the detritus mud (high-N substrate) from the rewetted fen Stangenhagen. Litterbags were placed in the containers in the anoxic zone of the substrate close to the bottom. (b) Litterbags with leaf and rhizome litter before incubation.

 $CO_2$ , or  $CH_4$  in some anoxic soils or its stabilization in other soils. The use of rather labile fresh litter was needed to study these processes over a short experimental time. Thus, our experiment can possibly be compared to reciprocal peat core or litter transplantation studies which similarly try to investigate the basic mechanisms of C and N cycling in specific ecosystems through modifications like placing high-N litter in a low-N environment.

Yet, it is true that most studies on N cycling focus on aerobic environments, e.g. leaf litter in mineral soils. So the question whether the analytical method can be applied to aerobic environments and whether similar effects can be expected will be of interest for many readers. We tend to believe that this is the case, but additional experiments are needed to affirm this. We would like to note that even for past decomposition studies, our analytical approach might be applicable if litter samples are
80 still available, stored under cold and dry conditions.

We will change the "Conclusions" of the manuscript to "Conclusion and perspectives":

**4 Conclusion and perspectives**

75

- In wetland ecosystems, the disentangling of gross N transformations is central for the assessment of C cycling, N cycling, and biogeochemical responses to increased external nutrient inputs, especially of N (Van Groenigen et al., 2015). Here, we have presented a new methodical approach which enables the disentangling of plant and microbial N in decomposed litter by using DNA signals in FTIR spectra as a marker for microbial N. This approach allows to quantify how much microbial N has formed and how much plant N is remaining in litter at a certain stage of decomposition.We have demonstrated that substrate-dependent, i.e. decomposition site dependent, variations in litter N accumulation were not caused by variations in
- 90 the amount of existing microbial N in litter at a certain stage of decomposition, but instead by variations in the remaining amount of still unprocessed plant-N. This indicates a decrease of gross protein depolymerization when litter decomposes in a nutrient-rich environment, and suggests that microorganims preferentially use inorganic N from porewater instead of organic N from litter when those two pools compete as consumable N sources. The influence of the decomposition site and its nutrient status on gross protein depolymerization in decomposing vascular litter has not been detectable in former studies. Instead, net
- 95 N mineralization/immobilization patterns in decomposing litter are often perceived as being predetermined solely by the initial litter C/N ratio and the rate of microbial N accumulation.

For litter decomposition in *Sphagnum* peat, we found high bulk N losses from litter due to preferential protein depolymerization. Although not directly adressed in this study, we assume that sphagnan, a compound released from *Sphagnum* mosses, is responsible for this effect. Sphagnan is known to bind free amino groups, so that a fraction of the depolymerized amino

- 100 acids might become inaccessible for microbial decomposers. We suggest that preferential protein depolymerization might be an adaptive mechanism to compensate amino acid losses to sphagnan. Preferential protein depolymerization leads to a gradual N depletion of the unprocessed organic matter in litter, what can be considered as a gradual decrease of litter decomposability, a potential mechanism for long-term preservation of vascular litter in *Sphagnum* peat.
- Litter decomposition has here been studied in anoxic and waterlogged wetland soils. While initial litter C/N ratios are usually successful predictors of N mineralization patterns in aerobic environments, e.g. mineral soils, we here found bulk litter N dynamics that did not fit into that established concept. The strength of the site-dependency of C/N-patterns in litter is propably caused by the waterlogged conditions. Nutrients can more easily be exchanged between litter and porewater compared to less humid mineral soils. Additionally, microbial nutrient demand is lower in anoxic environments why we were able to test a large gradient in initial litter N content without causing N limitation.
- 110 Yet, the here shown ability of microbial decomposers to preferentially release N from decomposing plant tissue is probably not limited to anoxic systems. In particular for the aerobic decomposition of low N litter, which in some cases has been shown to proceed without net N immobilization, preferential protein depolymerization is an alternative mechanism to the concurrently assumed lowered microbial CUE through overflow respiration. Microbial N in litter, determined through the here reported FTIR method, might be an easily available parameter to investigate trends in preferential protein depolymerization and CUE
- 115 in future aerobic litterbag studies.

Small method modifications should be considered for the applicability of the method in aerobic decomposition studies. These modifications include an optimization of the calibration curve, either through the addition of very low-N litters as calibration samples to a decomposition study or through the artificial mixing of undecomposed litter with microbial biomass. Furthermore, the contribution of fungi must be considered, which we assumed to be absent in anoxic soils. Differences in the amont

120 of DNA per biomass units and in the C/N ratio should be considered for the decomposer biomass in aerobic systems. Finally, the applicability of the same calibration curve for decomposed litters of different plant species still has to be investigated.

Table 1. I was a bit confused by \* decomposition in home soil. I would have thought the 'home soil' would be high-N leaf with high-N substrate, medium with medium etc., but this is not the pattern observed. I wondered why.

To answer your question we would like to present some data of a study we did earlier which includes the same leaf litter used in the decomposition experiment. In that preliminary study we compared the chemical composition of *P. austrails* leaf litters of wetlands with different nutrient status. The data are presented in Table 1 and Table 2. The decision on the sampling sites in the decomposition study was actually based on data of Table 2.

125

- 130 The expected correlation between soil N and leaf litter N is partly observed. Leaf litter from the mesotrophic wetlands of group 3 have nitrogen contents below 1.2%. The eutrophic wetlands of group 1 and 2 have leaf litter N contents ranging from 1.5 to 2.1%. In our study, leaf litter from the medium-N site has a lower N content than the litter from the high-N site, as expected. The acidic kettle mires, which all have peat moss floating mats and small populations of *P. australis*, fall out of this pattern. *P. australis* litter from these nutrient poor sites of group 4 have comparably high litter N contents. In our experiment, the most
- 135 nutrient rich litter belongs to the most nurtient poor site, which is the kettle mire Kablow Ziegelei. In the textbook by Reddy and DeLaune (2008, p. 315) it is written that "nitrogen content of plant tissue is inversely related to biomass. It is expected for plants with high biomass production that concentration of nitrogen may be lower as a result of dilution and distribution within the tissue." Even though Reddy and DeLaune's statement is based on data from McJannet et al. (1995), who compare different plant species, we believe that a similar mechanism is responsible for the high N content in *P*.
- 140 *australis* litter of group 4 sites. At these sites the *P. australis* plants were much smaller (shoot size and leaf area) compared to litter from more nutrient rich sites.

Another aspect which we would like to mention is the effect of silica. The ash content of the high-N leaf litter is only about 3%, while it is 7 and 10% for the other two leaf litters. The infrared spectrum of the high-N litter, shown in red in Figure 2a, shows a very low absorption intensity in the carbohydrate region. The black line is a difference spectrum which shows that a missing

- 145 peak at  $1095 \text{ cm}^{-1}$  causes this difference. In combination with bands at 800 and  $470 \text{ cm}^{-1}$  this peak is identified as biogenic silica (Figure 2b). The lack of silica in the leaves from the poor *Sphagnum* fen is also reflected by the C content, which is about 48% vs. 44% in leaves from the nutrient-richer sites. In line with a higher C content, the N content of the high-N leaf tissue could be expected to be about 10% higher at a similar organic matter composition, simply because the tissue is less diluted with inorganic constituents.
- 150 Yet, the C contents remained relatively stable throughout the decomposition experiment for all litters (Table S3 in the SI), so changing concentrations of inorganic constituents did not cause changes in N content in our study. Furthermore, conclusions in the manuscript are mostly based on C/N ratios and percentual C and N losses, which are invariable to the ash content.

**Table 1.** The twelve analysed Brandenburgian peatlands. Hydrogenetic mire type classification was performed according to Succow and Joosten (2001). The peatlands are divided into four groups with respect to nutrient levels. The three peatlands in the first group have been drained intensively in the past and have been rewetted in recent years resulting in the formation of flooded mires and new ecosystems.

|   | Sampling Site            | Hydrogenetic mire type  | Ecological mire type   | pН  | Latitude, Longitude    |  |
|---|--------------------------|-------------------------|------------------------|-----|------------------------|--|
|   |                          |                         |                        |     |                        |  |
|   | Moor bei Stangenhagen    | Flood mire              | euthrophic             | 6.2 | 52.204294°, 13.091846° |  |
| 1 | Moor bei Menzlin         | percolation mire        | euthrophic             | 6.6 | 53.88333°, 13.63333°,  |  |
|   | Niedermoor Hasenfelde    | Terrestrialisation mire | euthrophic             | 7.2 | 52.68333°, 14.38333°   |  |
|   | Moor bei Anklam          | Percolation Mire        | euthrophic             | 6.9 | 53.79599°, 13.83202°   |  |
| 2 | Glieningmoor             | Terrestrialisation mire | euthrophic             | 6.2 | 52.349555°, 14.199986° |  |
|   | Maxsee Niederung         | Percolation mire        | euthrophic             | 7.0 | 52.46308°, 13.97963°   |  |
|   | Töpchin Süd              | Terrestrialisation mire | mesotrophic calcareous | 6.6 | 52.161700°, 13.577400° |  |
| 3 | Triebschmoor             | Terrestrialisation mire | mesotrophic calcareous | 7.0 | 52.3454°, 13.80413°    |  |
|   | Pätzer Hintersee         | Terrestrialisation mire | mesotrophic subneutral | 6.3 | 52.20639°, 13.632832°  |  |
|   | Dollgengrund             | Terrestrialisation mire | mesotrophic subneutral | 5.7 | 52.00000°, 14.03333°   |  |
| 4 | Moor bei Kablow Ziegelei | Kettle mire             | oligotrophic acid      | 4.4 | 52.32571°, 13.72182°   |  |
|   | Kleiner Milasee          | Kettle mire             | oligotrophic acid      | 4.2 | 52.153220°, 13.957115° |  |

Stangenhagen, Menzlin and Hasenfelde (group 1) have been drained intensively in the past decades. In recent years, these sites have been rewetted and can now be classified as flooded mires in an early regeneration state. Anklam, Glieningmoor and Maxsee Niederung (group 2) are ground water fed, nutrient-rich mires in natural state. Töpchin Süd, Triebschmoor and Pätzer Hintersee (group 3) are nutrient-poor mires in near pristine condition, while Dollgen, Kablow-Ziegelei and Kleiner Milasee (group 4) are nutrient poor, acidic pristine mires.

Moor bei Stangenhagen: high-N substrate, Töpchin Süd: medium-N substrate, Moor bei Kablow Ziegelei: low-N substrate.

|   | Sampling Site           | С             | Ν              | C:N          | Р             | C:P          | Polyphenols    | Ash              |
|---|-------------------------|---------------|----------------|--------------|---------------|--------------|----------------|------------------|
|   |                         | (mg/g)        | (mg/g)         | (mass)       | (mg/g)        | (mass)       | (mg/g)         | (mg/g)           |
|   | Moor bei Stangenhagen   | $437.5\pm1.8$ | $14.6\pm0.1$   | $29.9\pm0.1$ | $0.81\pm0.01$ | $54.0\pm0.9$ | $18.4\pm2.0$   | $103.3\!\pm1.7$  |
| 1 | Moor bei Menzlin        | $454.1\pm0.2$ | $20.5\pm0.1$   | $22.1\pm0.1$ | $1.40\pm0.03$ | $32.4\pm0.9$ | $17.4\pm3.1$   | $58.4{\pm}1.4$   |
|   | Niedermoor Hasenfelde   | $464.1\pm2.1$ | $18.1\pm0.4$   | $25.6\pm0.4$ | $1.58\pm0.01$ | $29.4\pm0.3$ | $27.1\pm1.1$   | $46.8 {\pm} 0.9$ |
|   | Moor bei Anklam         | $464.3\pm0.2$ | $18.7\pm0.3$   | $24.9\pm0.5$ | $1.29\pm0.01$ | $25.9\pm0.1$ | $18.4\pm2.6$   | $50.6 \pm 1.2$   |
| 2 | Glieningmoor            | $471.2\pm0.9$ | $13.7\pm0.1$   | $34.5\pm0.3$ | $0.59\pm0.01$ | $79.3\pm0.1$ | $20.2\pm4.0$   | $43.7{\pm}3.4$   |
|   | Maxsee Niederung        | $432.3\pm1.1$ | $14.7\pm0.2$   | $29.4\pm0.3$ | $0.82\pm0.01$ | $53.0\pm0.9$ | $17.4\pm3.0$   | $93.3\!\pm1.4$   |
|   | Töpchin Süd             | $447.9\pm0.8$ | $9.2\pm0.1$    | $48.6\pm0.1$ | $0.65\pm0.02$ | $69.1\pm2.0$ | $11.9 \pm 1.4$ | $73.6{\pm}1.8$   |
| 3 | Triebschmoor            | $410.2\pm0.1$ | $11.7\pm0.1$   | $35.0\pm0.5$ | $0.50\pm0.01$ | $81.6\pm2.1$ | $19.0{\pm}2.1$ | $138.2{\pm}1.5$  |
|   | Pätzer Hintersee        | $442.7\pm3.5$ | $11.8\pm0.1$   | $37.4\pm0.2$ | $0.65\pm0.02$ | $68.2\pm1.2$ | $10.9{\pm}1.2$ | $77.1{\pm}2.3$   |
|   | Dollgengrund            | $461.8\pm0.4$ | $17,\!2\pm0.1$ | $26.9\pm0.1$ | $0.89\pm0.01$ | $51.6\pm0.3$ | $21.6{\pm}3.7$ | $60.4 {\pm} 0.4$ |
| 4 | Moor bei Kabow Ziegelei | $480.0\pm1.7$ | $20.3\pm0.4$   | $23.7\pm0.4$ | $0.90\pm0.01$ | $59.6\pm0.4$ | $32.4 \pm 4.4$ | $31.6{\pm}2.4$   |
|   | Kleiner Milasee         | $486.2\pm0.3$ | $20.6\pm0.1$   | $23.6\pm0.1$ | $0.70\pm0.01$ | $69.5\pm0.8$ | $26.9{\pm}2.6$ | $34.1\!\pm1.8$   |

Table 2. Elemental composition and polyphenol contents of P. australis leaf litter from 12 fens of different nutrient status.

Leaf litters were collected from north-east German peatlands during December 2012. Polyphenol content was determined following the Folin-Ciocalteu Assay. Ash content is the remaining mass after 3 h combustion at 450°C. Leaf litter was not preleached before analysis, so its elemental composition differs slightly compared to data in the main manuscript.

Figure 2. (a) Infrared spectra of undecomposed *P. australis* leaf litter. In black the difference spectrum "medium-N litter minus high-N litter" that extracts the peak which causes the drop in absorption intensity of the high-N litter in the carbohydrate region. (b) The spectrum of biogenic silica (from Meyer-Jacob et al., 2014).

7, 4 Litter bag experiments usually entail the early stages of decomposition, in this case 21 to 45% over 75 days. One wonders what the patterns may have been if the study allowed sampling earlier and later: in other words, are the processes identified here time-dependent in the decomposition path?

155

160

The main process which we identified is the preferential depolymerization of proteins in leaf litter as a parameter which seems to be characteristic for a specific decomposition site. This is a so far mostly neglected process. For example, it is written in Wikipedia, that "whether nitrogen mineralizes or immobilizes depends on the carbon-to-nitrogen ratio (C:N ratio) of the decomposing organic matter" (Wikipedia, n.d.). The basis for the discovery of this relationship is stoichiometric considerations which revealed a "global stoichiometry of litter nitrogen mineralization" (Manzoni et al., 2008) and the initial litter C/N ratio is identified as a predictor wether N mineralization or immobilization dominates.

We only have one sampling time (after 75 days), but large differences in overall C losses, ranging from 21 to 45%. Yet, many stoichiometric models which are the basis for some conceptual views on N cycling in decomposing litter do not use decompositon time as a parameter (Manzoni, 2017). N dynamics in litter are seen as a function of C loss. According the stoichiometric

165 models, this could be seen as a gradient in the decomposition path and within these concepts we think that there might be no specific time-dependent of these processes.

Still it would be worth to discuss a potential outcome if different sampling times were allowed. We believe that the first few decomposition days will probably be governed by leaching processes. Even when preleached leaf litter is used like in our study,
an initial (probably abiotic) increase of the C/N ratio is often observed in this phase. Stoichiometric models often exclude this phase and start which pre-decomposed litter. This effect of abiotic leaching is however not observable in our study, because abiotic N losses should be litter specific and independent of the porewater chemistry of a specific site.

For very long decomposition times, other conceptual assumptions might need a re-evaluation. Our study is somehow caught between the chairs. We started with the approach by Tremblay and Benner (2006) who quantified the amount of microbial
biomass and microbial N in decomposed litter using different markers. While Tremblay and Benner (2006) stopped at this point, we went one step further and use plant N and microbial N data in decomposed litter to caclulate CUEs and nNUEs. For this we use equations from isotope tracer experiments, which directly relate consumption and excretion. But tracer experiments

cover very short experiment times, so active elemental fluxes are compared. We consequently used a simplified model which assumes that plant organic matter breakdown and microbial biomass growth are the only two processes involved, what is not

180 realistic over the full decomposition path. In our picture the final outcome of litter decomposition would be no remaining plant OM and newly formed microbial biomass, whose amount corresponds to about 10% of the initial plant C. In nature, close to no organic matter remains at the end of decomposition.

In stoichiometric models, the underlying equation for C loss is

$$\frac{dC(t)}{dt} = -D + D \cdot CUE \cdot \frac{C}{C_0} \tag{1}$$

185 where D is the decomposition rate in C units (Manzoni, 2017). We ignored the term  $C/C_0$ . Stoichiometric models use the litter bulk C and N, and a decomposed litter sample that reached a high N content due to microbial biomass accumulation will not be different to a litter sample which comes with a high initial plant N content.

While we believe that for 75 days our methods are suitable, these might need to be adapted for long decomposition times. We think that the DNA band, which we quantified through FTIR, is a marker for the living microbial biomass in litter. At the end

- 190 of decomposition, we would expect that non-microbial C and N fractions in litter are not only plant OM, but a mixture of plant OM and remains of dead microbial biomass. A decrease of the (what we termed) plant OM C/N ratio could thus be expected over time. But such hypothesis would need more experiments. For our experiment, which lasted 75 days, we believe that such secondary processes like the cycling of microbial necromass can still be neglected.
- 195 We actually have performed another decomposition experiment with different sampling times. The aim of the study was to investigate how the breakdown of plant organic matter and the formation of DOM differs under different redox conditions. The experimental setup included leaf litter (from the rewetted fen Stangenhagen) with an inoculum in carbonate-buffered pure water (Figure 3). As N cycling was not the primary aim of that experiment, we would like to present here how our method performs over time (Table 3).
- 200 Without any natural environment and with the leaf litter

---

## Referee Comment (RC2) · Tim Moore (Referee) · 23 Jul 2019

Thank you for your informative response to the points that I raised. It adds to the 'value' of your results and the context, without 'drowning' the manuscript in ancillary information which would distract/deter the reader and make it a long treatise. Tim Moore

---

## Referee Comment (RC3) · Anonymous Referee #2 · 29 Sep 2019

The main aim of the study was to evaluate the fate of plant litter nitrogen in a decomposition experiment involving litters and peaty soils with contrasting N status. It was necessary to distinguish between two fractions of protein nitrogen in the litter: (1) remaining original N that has not been depolymerized by decomposers' enzymes and (2) newly synthesized microbial N. The authors proposed a novel approach how to distinguish the two fractions; they measured precise FTIR spectra to evaluate peaks of total protein nitrogen and microbial DNA phosphorus (assuming that the DNA P is associated only with the microbes). Assuming constant microbial N:P stoichiometry they could express the microbial N fraction. I am not a microbiologist, so I am not able to review critically the assumptions leading up to the evaluation of preferential protein

depolymerization. However, I appreciate the careful explanation of all the evaluation steps supported by references. The manuscript is well and clearly written but I would like to discuss following issues.

How is the microbial N invested in extracellular enzymes accounted for? How relevant is this fraction in the evaluation of the N fate in the decomposing litter? How can it differ in N-poor/rich soils and litters? How this fraction can affect the proposed method leading to the evaluation of preferential protein depolymerization?

The concept also does not mention that the extracellular enzymes may mediate N acquisition from dissolved organic N, which was not analyzed in the soil water. How relevant is this N pool in the tested soils?

Why anoxic conditions were chosen for the experiment? Most plant litters, also in peatland habitats, are first exposed to oxic conditions. Do you think the conclusions are fully applicable also in oxic decomposition where fungal decomposition often prevails?

Other comments: Chapter 2.3 Infrared Spectroscopy: I think that more details about the target compounds and their absorption bands can be provided here in the method description than only later in the Results and Discussion.

P4, L21: How effective was the 17-h period in leaching the litter? Is it possible that a significant proportion of the mass loss can be still attributed to the leaching and not entirely to microbial activity?

P4, L23: How was the rhizome litter defined? (I expect that a continuum between living and highly decomposed rhizomes can be found in the soil).

P12, L12 and Figure 4b: The linear model has the intercept very close to zero (as indicated by the trendline in the graph), obviously statistically not different from zero. What is the relevance of the zero intercept? Does it support the assumption of the entirely microbial origin of the rhizome litter N and P?

Technical comments: Table 1, first column: "soil substrate" can be clearer (as it is used

also in the text)

P4, L4: "N mineralization/immobilization": does the slash sign denote a ratio or something like "and/or"?

P4, L12: Replace "sedge-brown moss peat" by "sedge–brown-moss peat"

P7, L1: The C/N in the senescent leaves was measured after the leaching? If so, "leached leaves" can be used.

P19, L12: Although the data on CuO-oxidation lignin monomer products were not used in the paper, the supplement should contain a description of the method (or a reference).

---

## Author Comment (AC2) · 10 Nov 2019

**Reviewer comment: The main aim of the study was to evaluate the fate of plant litter nitrogen in a decomposition experiment involving litters and peaty soils with contrasting N status. It was necessary to distinguish between two fractions of protein nitrogen in the litter: (1) remaining original N that has not been depolymerized by decomposers' enzymes and (2) newly synthesized microbial N. The authors proposed a novel approach how to distinguish the two fractions; they measured precise FTIR spectra to evaluate peaks of total protein nitrogen and microbial DNA phosphorus (assuming that the DNA P is associated only**

**with the microbes). Assuming constant microbial N:P stoichiometry they could express the microbial N fraction. I am not a microbiologist, so I am not able to review critically the assumptions leading up to the evaluation of preferential protein depolymerization. However, I appreciate the careful explanation of all the evaluation steps supported by references. The manuscript is well and clearly written but I would like to discuss following issues.**

Authors reply: We would like to thank you for taking the time to review our manuscript. We furthermore thank you for the concise summary of our study and for your helpful and constructive comments which will improve the manuscript. Below we will respond to all issues raised and indicate how it was assisting for the revision of the manuscript accordingly.

**[1] How is the microbial N invested in extracellular enzymes accounted for? [2] How relevant is this fraction in the evaluation of the N fate in the decomposing litter? [3] How can it differ in N-poor/rich soils and litters? [4] How this fraction can affect the proposed method leading to the evaluation of preferential protein depolymerization?**

[1] Our methodological approach taken is only capable of quantifying microbial N from the FTIR peak heights of DNA bands. Plant bound N is determined as total litter N minus microbial N. The extracellular enzyme N-pool, located outside the microbial cells, yet part of the microbial N-fraction, is not considered as a standalone paramter.
[2] Unfortunately, we were not able to find quantitative data on the importance of this N-pool in the scientific literature. Enzymatic studies commonly do not quantify this pool directly but measure enzyme activities. This can partly be rationalized by a lack of efficient exoenzyme extraction methods that avoid cell lysis and a co-extraction of cell-bound enzymes.
Yet, we assume that the exoenzyme N-pool will be small and negligible compared to plant and microbial N. This can be illustrated indirectly as a fraction of DON in soil porewater should be composed of microbial exoenzymes. The detritus mud in this

study can be considered as an environment with high microbial activity in which we measured about 10 mg/L DON. If we assume that 1 g leaf litter is soaked with 2 mL water which contains 10 mg/L DON (a fraction of which is the exoenzymes), the DON would only sum up to 20 $\mu$g water-soluble N per gram leaf litter, a rather small amount compared to the 10-30 mg total (insoluble) N per gram litter.

[3] In soils, especially mineral soils, the "exoenzyme N-pool" can account for a high fraction of the total N because exoenzymes can be immobilized and stabilized on mineral surfaces. Within leaf litter, this immobilization process should be less relevant and the exoenzymes themself will in all systems account for a very small fraction of the total organic N.

The interpretation of C/N-changes in decomposing litter using the stoichiometric decomposition theory acknowledges the effects of C-limitation or N-limitation as a function of soil-N or litter-N. But this is mostly discussed in terms of microbial biomass growth and decomposition activity, without explicitly mentioning variations in the extracellular enzyme release rate.

[4] As discussed in the manuscript, there were no indicators for N-limitation in any decomposed leaf sample. The CUEs were very similar for all litters what indicated a similar microbial biomass growth over the decomposition path within all samples. This uniformity should also apply to the exoenzyme production and release rate.

It can be supposed that the preferential depolymerization of proteins over other plant biopolymers, as observed in the poor *Sphagnum* peat soil, is induced by a higher formation and release of protease, the protein depolymerizing exoenzyme. Yet, this will only very minorly affect the overall exoenzyme N-pool because protease is only one enzyme out of many different enzymes required to depolymerize the complex plant tissue.

Changes in manuscript: Exoenzymes are along with the readily depolymerized amino acids a small and rapidly cycling water-soluble N-pool in decomposing litter which is commonly not quantitatively discussed in studies on C/N dynamics in litter. We, therefore, decided not to discuss this pool explicitly, but only to mention the two

N-pools in litter which are plant N and microbial N. Yet, we will mention exoenzymes as part of DON which will receive more attention in the revised manuscript.

**The concept also does not mention that the extracellular enzymes may mediate N acquisition from dissolved organic N, which was not analyzed in the soil water. How relevant is this N pool in the tested soils?**

DON was quantified as $0.82 \pm 0.19$ mg/L in the low-N substrate, $0.73 \pm 0.16$ mg/L in the medium-N substrate, and $8.12 \pm 0.77$ mg/L in the high-N substrate. N acquisition from DON might thus be a relevant process in the high-N environment, a process that we overlooked in the present manuscript. We thank reviewer 2 for this comment and will discuss this process in the revised manuscript.
In our study, the potential effect of DON parallels that of porewater ammonium. DON and ammonium are external N sources available to microbes. Furthermore, both external N sources are only available in the high-N substrate, for which effects of ammonium are discussed. A consideration of DON along with DIN can thus easily be implemented.

Changes in manuscript: We will include DON along with ammonium as an additional external N source.

**Why anoxic conditions were chosen for the experiment? Most plant litters, also in peatland habitats, are first exposed to oxic conditions. Do you think the conclusions are fully applicable also in oxic decomposition where fungal decomposition often prevails?**

It is true that above-ground leaf litter in peatland habitats initially decomposes under oxic conditions. However, if peatlands are inundated as it is the case after rewetting of degraded peatlands, the plant litter might reach the low-oxygen or even anaerobic zone of the detritus layer within few days only and decompose anaerobically at an early stage of decomposition. Anaerobiosis in this newly formed mud layer might cause carbon sequestration which is well documented (Cabezas et al. 2014). On the

other hand, anoxic decomposition can be very efficient in anoxic environments like rewetted fens, indicated by the high $CO_2$ and $CH_4$ release from these ecosystems. The elucidation of carbon preservation mechanisms in anoxic environments is of high importance. We likewise chose anoxic decomposition as we had noticed, that anoxic litter decomposition in peatlands had only limitedly been studied using a litterbag experimental design.

We tend to believe that the capability of microbes to adjust their protein depolymerization activity in response to external N availability is not limited to anoxic soils, at least when litter decomposes subaqueously or under very humid conditions where diffusion of porewater N is high so that microorganisms can easily access external N sources. Yet, further experiments are needed to test these assumptions. The described analytical approach should generally apply to litterbag studies in oxic environments, but a different microbial biomass C/N ratio and a different DNA:N ratio must be expected when fungal decomposition prevails.

Changes in manuscript: We will change the "Conclusions" of the manuscript to "Conclusions and perspectives" in order to discuss the potential outcome of the reported method in oxic decomposition studies. The revised version of the conclusions can be found in the answer to Reviewer 1.

**Other comments: Chapter 2.3 Infrared Spectroscopy: I think that more details about the target compounds and their absorption bands can be provided here in the method description than only later in the Results and Discussion.**

We agree and will add information on how quantitative data for amide I and amide II bands as indicators for litter protein content and semiquantitative data for DNA bands were extracted from the FTIR spectra.

**P4, L21: How effective was the 17-h period in leaching the litter? Is it possible that a significant proportion of the mass loss can be still attributed to the leaching and not entirely to microbial activity?**

We did not collect data on mass loss or DOC during the leaching procedure, so we cannot precisely answer your first question. However, it can be assumed that first leaching is the dominant process and that with increasing time the microbial decomposition becomes the dominant process indeed coupled partly with ongoing leaching processes (Asaeda et al. 2002). It must be noted that leaching was already taking place under in-situ conditions. The weeks preceding the leaf litter sampling were rather rainy, so leaching by rain-water will already have occurred in the field. The leaching in the lab was done to remove water-soluble organic matter, but also to reduce effects of the inhomogeneous nature of the natural leaching by rain between leaf parts (leaf top vs. leaf bottom, etc.) as well as between sampling sites.

We have data of a leaching experiment for which the leaf litter from the kettle-mire Kablow-Ziegelei was used. 4.5 g leaf litter was leached in 1 L water for 24 h. DOC (0.2 $\mu$m filtered) reached 92 mg/L what corresponds to 4.4% of the initial leaf C.

Even though all leaves were pre-leached, significant amounts of organic matter were likely lost from the litter by abiotic leaching during the 75 days. Yet, such a leaching effect should be litter specific, unaffected by differences in water chemistry of the three decomposition substrates. Theoretically, the low C-losses of the medium-N leaf litter in all substrates could be caused by a lower extent of leaching compared to the other two leaf litters. Yet, the low amounts of microbial N in these samples indicate a lower microbial decomposability of this sample as a cause of the low C-loss. The most important findings in this study, which are the site-specific differences in the decomposition process for each litter type (preferential protein depolymerization), cannot be explained by litter-specific leaching effects.

Changes in manuscript: As we did not collect data on the leaching process we decided not to mention the leaching in more detail in the manuscript.

**P4, L23: How was the rhizome litter defined? (I expect that a continuum between living and highly decomposed rhizomes can be found in the soil).**

The rhizome litter was cut from living *P. australis plants*, rinsed with distilled water and

freeze-dried. We thank you for this comment, as we forgot to mention, that the rhizome tissue was indeed living plant tissue. We became aware that the term "litter" does not strictly apply to belowground living plant tissue and will change the term "rhizome litter" to "rhizomes" or "rhizome tissue" in the manuscript.

**P12, L12 and Figure 4b: The linear model has the intercept very close to zero (as indicated by the trendline in the graph), obviously statistically not different from zero. What is the relevance of the zero intercept? Does it support the assumption of the entirely microbial origin of the rhizome litter N and P?**

The trend line in Figure 4b indeed has an intercept very close to zero. The exact formula of the trendline presented in Figure 4b in the manuscript is: y= 0.05996 + 0.29411x.
Indeed, the trendline shown in the original Figure 4b was not statistically different from zero. To account for the DNA:N homeostasis, we used a trendline forced through zero. The formula of that trendline was y=0.31452x, as denoted in the manuscript. We will replace Figure 4b with the used trendline intercepting zero. The novel Figure 4b with the changed trendline has been added to the end of this document.
The finding, that N values and DNA signals lead a trendline very close to zero for the rhizomes was indeed an important finding during the method development. At that time, we tried to find an overall pattern in the complex and incomprehensible dataset of litter C/N changes. We were aware that only very little plant N could remain within the decomposed rhizomes while leaf litter N had to be a mixture of plant and microbial N. Assuming microbial homeostasis, we were searching for a potential marker of microbial N in the litter using infrared spectroscopy. The DNA bands were a promising start, but in the scientific literature, these had only been reported in the spectra of microbial cell tissue. The attempt to extract these bands by using the second derivative spectra of FTIR difference spectra was an experimental approach that could easily have reached the limits of infrared spectroscopy in terms of resolving power and signal-to-noise ratio. Yet, we found a positive signal for all decomposed

litter samples at the position where DNA-band should appear. The N-to-DNA trend line of the rhizomes, closely passing the origin, was an important confirmation that we did extract a DNA signal as this pattern was expected from stoichiometric considerations. Furthermore, it allowed quantifying microbial N in leaf litter samples. This quantification of microbial N ultimately allowed the calculation of CUE-values which were very constant for all leaves, what again was a pattern consistent with the stoichiometric decomposition theory. Only then we were rather confident that we positively extracted DNA signals and that the site-dependent deviations in N dynamics were due to variations in remaining plant N what indicated site-dependent preferential protein depolymerization.

Changes in the manuscript: We will replace Figure 4b (new version shown at the end of this document) and put some more emphasis on the importance of the trend line closely passing zero.

**Technical comments: Table 1, first column: "soil substrate" can be clearer (as it is used also in the text)**

We agree and will use the term "organic soil" instead of "substrate".

**P4, L4: "N mineralization/immobilization": does the slash sign denote a ratio or some- thing like "and/or"?**

We will change the term "net N mineralization/immobilization" to "net N mineralization and/or immobilization". We thank you for bringing this imprecision to our attention.

**P4, L12: Replace "sedge-brown moss peat" by "sedge–brown-moss peat"**

It will be done.

**P7, L1: The C/N in the senescent leaves was measured after the leaching? If so, "leached leaves" can be used.**

We thank you for this comment and will use "leached leaves".

**P19, L12: Although the data on CuO-oxidation lignin monomer products were not used in the paper, the supplement should contain a description of the method (or a reference).**

We agree and will add a short paragraph on lignin analysis in chapter 2.2, mentioning that CuO lignin data of the organic soils can be found in the supplement information and lignin data of leaf litters in the supplementary datasets.

Thank you again for your review,
Hendrik Reuter, on behalf of all coauthors

**References:**

Asaeda, T., Nam, L. H., Hietz, P., Tanaka, N., Karunaratne, S.: Seasonal fluctuations in live and dead biomass of *Phragmites australis* as described by a growth and decomposition model: implications of duration of aerobic conditions for litter mineralization and sedimentation, Aquatic Botany, 73, 223-239, 2002.

Cabezas, A., Pallasch, M., Schönfelder, I., Gelbrecht, J., Zak, D.: Carbon, nitrogen, and phosphorus accumulation in novel ecosystems: Shallow lakes in degraded fen areas, Ecological Engineering, 66, 63-71, 2014.

[Figure]

[Figure]

**Fig. 1.** Novel Figure 4b with the trendline forced through the origin.

---

## Author Comment (AC3) · 10 Nov 2019

Dear Tim Moore,
we thank you again for the review of our work, which has given room for a more in-depth
discussion of some of the study's findings. We also thank you for your agreement not
to include all additional information in the main document.
All the best,
Hendrik Reuter on behalf of all co-authors

―――――――――――――――――――

---

## Author Response (AR1)

**Manuscript changes in response to Reviewer 1:**

All grammatical/typographical and stylistic errors identified by Reviewer 1 (as noted in the provided PDF-file) have been corrected.

Changes in Figures:

Figure 2 (new numbering): has been newly added to illustrate the experimental setup.

Figure 3: "Rhizome litter" has been replaced by "Rhizome tissue", a larger font size has been chosen for the legend.

Figure 5b: The trend line now passes zero, the $R^2$ has been adjusted accordingly.

**Reviewer Comment:**

**1, 0 While I think that the preservation of vascular litter in Sphagnum peat is a useful product of the work, I think it has a broader impact, and most litter entering Sphagnum (and other) peats decays initially under aerobic conditions, rather than the anaerobic burial used in this experiment. Thus I would suggest a more generic title, emphasizing the more original approaches taken.**

Changes made: We have changed the title to "Evidence for preferential protein depolymerization in wetland soils in response to external nitrogen availability provided by a novel FTIR routine".

**1, 2 What does 'relative' mean here? It could be N accumulation relative to N (implying a lowering of the C:N ratio) or it could be a larger N mass, relative to the initial litter. Please clarify.**

The beginning of the abstract has been changed (line 1 on page 1).

**4, 25 The experiment was conducted under anaerobic conditions, or at least litter placed in containers into which substrates had been added and presumably under saturated or waterlogged conditions. I think this is important, partly because of the conditions created (anoxic) and, as I note above, most peatland vascular litter does not decompose initially under anaerobic conditions. Thus, I think the experimental details of these containers and substrates/litter need to be better described. Also, were they incubated at 'room temperature'? Furthermore, are the results of this study likely to be repeated, quantitatively or qualitatively, if the experiment was to be repeated under aerobic conditions, which is probably the situation in many wetlands. Of course, one could argue that the initial aerobic decomposition is followed by anaerobic, as the litter becomes buried and goes beneath the water table.**

The term "room temperature" has been changed to $20 \pm 1°C$ (line 22 on page 4).

The decomposition experiment is now described in more detail in the "Materials and methods". Furthermore, an additional Figure has been added to this section which shows the soil containers and the litterbags.

The "conclusions" of the manuscript have been replaced by a subsection called "Conclusion and perspectives" which now includes a short discussion of the potential performance of our method under oxic decomposition conditions (line 31 on page 18).

**Table 1. I was a bit confused by * decomposition in home soil. I would have thought the 'home soil' would be high-N leaf with high-N substrate, medium with medium etc., but this is not the pattern observed. I wondered why.**

We have discussed this issue in the response to reviewer 1 and will cite the "interactive discussion"-document in the main manuscript (line 3 on page 8).

**7, 4 Litter bag experiments usually entail the early stages of decomposition, in this case 21 to 45% over 75 days. One wonders what the patterns may have been if the study allowed sampling earlier and later: in other words, are the processes identified here time-dependent in the decomposition path?**

That was an interesting question which we have discussed in depth in the interactive discussion. Yet, our study only targeted the early phase of decomposition why a time-dependency of the reported findings could only be discussed on a speculative basis. We thus decided not to include this aspect in the manuscript. No changes were made.

**7, 8 Litter quality involves several attributes of the initial litter influencing decomposition rate, of which the C:N ratio is frequently cited. It was not borne out here, possibly because decomposition was under anaerobic conditions. Were**

**there any other attributes of the litter which might explain this deviation, such as P content, lignin content etc.?**

Unfortunately, our data provide no explanation for the differences in litter quality why we are not able to answer this question. No changes were made in the revised manuscript.

**Table S1 While nitrate was essentially non-existent in the porewater from the three substrates, there was a major difference in NH4 and also DOC, the latter implying a large variation in dissolved organic nitrogen (DON), referred to p 18, l6. In Sphagnum peatlands, DON dominates the pore water, often forming 60-90% of the total dissolved nitrogen (TDN). It appears that TDN was not measured (allowing an estimate of DON) but could there be more consideration of DON in the understanding of the processes involved?**

We replaced the data of Table S1 in the supporting information with data of a field sampling campaign that include data for DON.
As reviewer 1 made us aware that DON dominates the porewater of *Sphagnum* peatlands, we included this fact in the discussion of the role of sphagnan (line 3 on page 18).
We added DON as a potential external N source in the introduction (line 6 on page 3).

**I found it a little bit confusing that C and N ratio was expressed atomically, whereas everything appears to be on a mass basis; while atomic units are common in stoichiometric studies, most decomposition studies use mass.**

We have replaced all atomic C/N ratios by mass C/N ratios throughout the manuscript.

**Sequence of reference citations seems to vary between alphabetical and chronological and the format used in the References is variable.**

We changed the order of references in the manuscript text to an alphabetical order. We furthermore corrected typographical inconsistencies in the reference list. Please note that this kind of changes cannot be visualized in LATEX.

**In case the Supplement does not load, oxycoccos is mis-spelt and it is Electrical conductivity.**

These errors have been corrected in the supporting information file which can be found at the end of this document.

**Manuscript changes in response to Reviewer 2:**

Reviewer comment: **How is the microbial N invested in extracellular enzymes accounted for? How relevant is this fraction in the evaluation of the N fate in the decomposing litter? How can it differ in N-poor/rich soils and litters? How this fraction can affect the proposed method leading to the evaluation of preferential protein depolymerization?**

Changes made: While this is an important aspect of litter decomposition, the presented method is unable to consider this N-fraction in particular. Due to the lack of data we decided not to include a discussion of this N-pool in the revised manuscript. No changes have been made.

**The concept also does not mention that the extracellular enzymes may mediate N acquisition from dissolved organic N, which was not analyzed in the soil water. How relevant is this N pool in the tested soils?**

Changes made: Dissolved organic nitrogen has reveived more attention in the revised manuscript. Furthermore, a dataset on porewater chemistry which includes DON in the analyzed wetlands has been included. Details on the changes in the revised manuscript are described in the answer to the corresponding question of reviewer 1.

**Why anoxic conditions were chosen for the experiment? Most plant litters, also in peatland habitats, are first exposed to oxic conditions. Do you think the conclusions are fully applicable also in oxic decomposition where fungal decomposition often prevails?**

Changes made: We have replaced the "Conclusions" subsection of the manuscript by a subsection called "Conclusion and

perspectives" which now covers a discussion of the potential existence of preferential protein depolymerization and of the potential applicability of our method in oxic decomposition environments (line 31 on page 18).

**Other comments: Chapter 2.3 Infrared Spectroscopy: I think that more details about the target compounds and their absorption bands can be provided here in the method description than only later in the Results and Discussion.**

Changes made: We fully agree and have added these information as proposed by reviewer 2 (line 19 on page 6).

**P4, L21: How effective was the 17-h period in leaching the litter? Is it possible that a significant proportion of the mass loss can be still attributed to the leaching and not entirely to microbial activity?**

Changes made: We unfortunatelly did not collect data on the leaching procedure. We discussed this issue in the interactive discussion where we argued that there are no indicators of leaching effects in our data, why we decided not to discuss this topic in more detail in the revised manuscript. No changes were made.

**P4, L23: How was the rhizome litter defined? (I expect that a continuum between living and highly decomposed rhizomes can be found in the soil).**

Changes made: We added the information that the rhizomes were harvested from living plants (line 4 on page 5). We replaced the term "rhizome litter" by "rhizome tissue" or "rhizomes" throughout the manuscript.

**P12, L12 and Figure 4b: The linear model has the intercept very close to zero (as indicated by the trendline in the graph), obviously statistically not different from zero. What is the relevance of the zero intercept? Does it support the assumption of the entirely microbial origin of the rhizome litter N and P?**

Changes made: We have replaced Figure 4b (now Figure 5b) which now shows the trendline passing through zero. We furthermore added a short discussion of the significance of the trendline (line 5 on page 12).

**Technical comments: Table 1, first column: "soil substrate" can be clearer (as it is used also in the text)**

Changes made: We replaced the term "soil substrate" by "organic soil" in Table 1 and by "soil" in Table 2.

**P4, L4: "N mineralization/immobilization": does the slash sign denote a ratio or something like "and/or"?**

Changes made: We removed the fragment "/immobilization" (line 8 on page 4).

**P4, L12: Replace "sedge-brown moss peat" by "sedge-–brown-moss peat"**

Changes made: This error has been corrected throughout the manuscript and in the supporting information.

**P7, L1: The C/N in the senescent leaves was measured after the leaching? If so, "leached leaves" can be used.**

Changes made: We replaced "Senescent *P. australis* leaves from these sites..." by "Leached *P. australis* leaf litters from these sites...".

**P19, L12: Although the data on CuO-oxidation lignin monomer products were not used in the paper, the supplement should contain a description of the method (or a reference).**

Changes made: We included a short paragraph on lignin analysis in the "Materials and methods" section (line 16 on page 5).

[revised manuscript text omitted]

\* Decomposition in home soil.

23.4 mg/L $NH_4^+$-N and 8.1 mg/L DON in the porewater. A sedge–brown-moss peat substrate had a C/N of 19.7, 2.1 mg/L $NH_4^+$-N and 0.7 mg/L DON, and a *Sphagnum* peat a C/N of 33.1, about 1.0 mg/L $NH_4^+$-N and 0.8 mg/L DON. Further site information is provided in Table S1. For convenience, we will term the three decomposition environments "high-N soil", "medium-N soil", and "low-N soil". Leached *P. australis* leaf litters from these sites had initial
* * *
[c41] 72
[c42] *Text added.*
[c43] sedge–brown moss peat
[c44] 23.1
[c45] and
[c46] 1.4
[c47] *Text added.*
[c48] 38.6
[c49] and
[c50] *Text added.*
[c51] substrate
[c52] substrate
[c53] substrate
[c54] Senescent
[c55] leaves

[revised manuscript text omitted]
 | 0.96 ± 0.03 | 0 | 0 | [c]45.7 ± 2.1 | 0 | – | – | 0.96 ± 0.03 | 0 | 0 | – | [d]45.7 ± 2.1 |
| high-N soil | 1.71 ± 0.03 | 44.0 ± 1.7 | 1.4 ± 3.6 | [e]25.9 ± 0.2 | 0.67 ± 0.02 | 8.77 ± 0.61 | 96.9 ± 9.1 | 1.04 ± 0.01 | 48.3 ± 1.5 | 40.0 ± 2.3 | 0.83 ± 0.02 | [f]39.3 ± 0.4 |
| medium-N soil | 1.67 ± 0.05 | 45.0 ± 2.4 | 3.8 ± 4.3 | [g]26.1 ± 0.7 | 0.72 ± 0.04 | 9.21 ± 1.13 | 92.0 ± 8.4 | 0.95 ± 0.08 | 49.6 ± 2.1 | 45.3 ± 4.3 | 0.92 ± 0.09 | [h]42.3 ± 3.2 |
| low-N soil | 1.18 ± 0.07 | 32.5 ± 2.0 | 16.7 ± 3.1 | [i]37.0 ± 2.2 | 0.46 ± 0.01 | 9.99 ± 0.92 | 66.5 ± 4.6 | 0.71 ± 0.06 | 36.1 ± 1.9 | 49.5 ± 3.3 | 1.38 ± 0.16 | [j]58.0 ± 5.3 |
| **— medium-N leaf litter —** | | | | | | | | | | | | |
| initial | 1.39 ± 0.07 | 0 | 0 | [k]30.4 ± 0.8 | 0 | – | – | 1.39 ± 0.06 | 0 | 0 | – | [l]30.4 ± 0.8 |
| high-N soil | 1.97 ± 0.13 | 33.9 ± 1.5 | 11.6 ± 4.5 | [m]22.8 ± 1.1 | 0.35 ± 0.03 | 7.02 ± 0.78 | 58.6 ± 9.9 | 1.63 ± 0.11 | 36.5 ± 1.4 | 27.1 ± 4.1 | 0.74 ± 0.11 | [n]26.5 ± 1.6 |
| medium-N soil | 1.84 ± 0.03 | 38.7 ± 0.5 | 20.3 ± 2.0 | [o]23.4 ± 0.4 | 0.41 ± 0.00 | 6.93 ± 0.19 | 46.5 ± 2.9 | 1.44 ± 0.03 | 41.5 ± 0.5 | 37.8 ± 1.8 | 0.91 ± 0.03 | [p]28.5 ± 0.6 |
| low-N soil | 1.20 ± 0.06 | 21.1 ± 0.9 | 31.8 ± 3.4 | [q]35.3 ± 2.0 | 0.16 ± 0.06 | 6.58 ± 1.94 | 22.0 ± 5.5 | 1.04 ± 0.09 | 22.6 ± 1.4 | 41.0 ± 5.2 | 1.82 ± 0.21 | [r]40.1 ± 3.2 |
| **— high-N leaf litter —** | | | | | | | | | | | | |
| initial | 2.14 ± 0.11 | 0 | 0 | [s]21.9 ± 1.6 | 0 | – | – | 2.14 ± 0.11 | 0 | 0 | – | [t]21.9 ± 1.6 |
| high-N substrate | 2.83 ± 0.09 | 40.2 ± 1.9 | 23.5 ± 1.6 | [u]17.0 ± 0.4 | 0.55 ± 0.06 | 7.81 ± 0.20 | 38.7 ± 1.9 | 2.28 ± 0.05 | 43.6 ± 2.1 | 38.3 ± 2.2 | 0.88 ± 0.02 | [v]19.9 ± 0.4 |
| medium-N substrate | 2.66 ± 0.16 | 43.0 ± 2.2 | 33.3 ± 0.7 | [w]18.6 ± 0.9 | 0.63 ± 0.05 | 7.80 ± 1.09 | 32.1 ± 2.1 | 2.04 ± 0.18 | 46.6 ± 1.9 | 49.0 ± 2.3 | 1.06 ± 0.09 | [x]22.9 ± 1.8 |
| low-N substrate | 2.15 ± 0.09 | 25.7 ± 3.4 | 27.0 ± 3.8 | [y]22.2 ± 0.8 | 0.36 ± 0.03 | 9.97 ± 1.13 | 31.5 ± 4.3 | 1.79 ± 0.06 | 28.5 ± 3.4 | 39.3 ± 3.4 | 1.39 ± 0.11 | [z]25.7 ± 0.9 |

[a] Decomposed litter contains plant and microbial N.

[b] $N_{microbial}$: N content in decomposed litter belonging to microorganisms (in wt% litter dry mass).

[c] CUE: Carbon use-efficiency.

[d] nNUE: Net nitrogen use-efficiency.

[e] $N_{plant}$: N content in decomposed litter belonging to plant OM (in wt% litter dry mass).

[f] $C_{depoly.}$: Fraction of initially present plant C which has been depolymerized.

[g] $N_{depoly.}$: Analogously to $C_{depoly.}$.

[h] ($\alpha$): Coefficient of preferential protein decomposition ($N_{depoly}/C_{depoly}$).

[i] $(C/N)_{plant}$: Carbon-to-nitrogen ratio of the plant OM fraction within the litter.

[a]

[b]

[c]

[d]

[e]

[f]

[g]

[h]

[i]

[j]

[k]

[l]

[m]

[n]

[o]

[p]

[q]

[r]

[revised manuscript text omitted]

---

## Author Response (AR2)

**Author's Response:**

Dear Steven,

thank you for accepting our manuscript for publication in *Biogeosciences*. Please be aware that we have added some additional words to the acknowledgements which now add contribution to the technical staff at IGB and to the colleagues involved in the peer review process.

We greatly appreciate your guidance throughout the review process and the detailed comments of the two reviewers which have greatly improved the manuscript.

Best regards,
Hendrik Reuter, on behalf of all co-authors